# Cellular and Molecular Mechanisms of Pathogenesis Underlying Inherited Retinal Dystrophies

**DOI:** 10.3390/biom13020271

**Published:** 2023-02-01

**Authors:** Andrew Manley, Bahar I. Meshkat, Monica M. Jablonski, T.J. Hollingsworth

**Affiliations:** 1Department of Ophthalmology, Hamilton Eye Institute, College of Medicine, University of Tennessee Health Science Center, Memphis, TN 38163, USA; 2Department of Physiology, University of Tennessee Health Science Center, Memphis, TN 38163, USA; 3Department of Anatomy and Neurobiology, University of Tennessee Health Science Center, Memphis, TN 38163, USA; 4Department of Pharmaceutical Sciences, University of Tennessee Health Science Center, Memphis, TN 38163, USA; 5Department of Genetics, Genomics and Informatics, University of Tennessee Health Science Center, Memphis, TN 38163, USA; 6Department of Microbiology, Immunology, and Biochemistry, University of Tennessee Health Science Center, Memphis, TN 38163, USA

**Keywords:** inherited retinal dystrophy, retinitis pigmentosa, leber congenital amaurosis, stargardt’s disease, rod-cone dystrophy, cone-rod dystrophy, molecular mechanisms of pathogenesis

## Abstract

Inherited retinal dystrophies (IRDs) are congenital retinal degenerative diseases that have various inheritance patterns, including dominant, recessive, X-linked, and mitochondrial. These diseases are most often the result of defects in rod and/or cone photoreceptor and retinal pigment epithelium function, development, or both. The genes associated with these diseases, when mutated, produce altered protein products that have downstream effects in pathways critical to vision, including phototransduction, the visual cycle, photoreceptor development, cellular respiration, and retinal homeostasis. The aim of this manuscript is to provide a comprehensive review of the underlying molecular mechanisms of pathogenesis of IRDs by delving into many of the genes associated with IRD development, their protein products, and the pathways interrupted by genetic mutation.

## 1. Introduction

Inherited retinal dystrophies (IRDs) is an umbrella term for a multitude of retinal neurodegenerative diseases that are the result of genetic mutation derived from a myriad of genes whose protein products are inherently associated with a critical factor of retinal function and support [1,2,3,4,5,6,7,8]. Some examples of these protein products are shown in Figure 1, including their localization in the mouse retina using immunohistochemical labeling. Interestingly, the majority of proteins associated with IRDs are localized to the same cell types, organelles, and structures in mice as they are in humans, making them an important model for studying these diseases. The two main categories of IRDs are rod-cone dystrophies (RCDs) and cone-rod dystrophies (CRDs), with each group characterized by a primary loss of one of the specific photoreceptor cell types—rods or cones—and a secondary loss of the other cell type [1,6,9,10,11,12,13]. It is important to note that, while IRDs primarily affect the photoreceptors’ survival, the mutated genes causing these diseases are not necessarily expressed by the photoreceptors. In fact, congenital retinal pigment epithelium (RPE) dysfunction is a major cause of both RCDs and CRDs and is detailed in Section 4 below. Examples of RCDs include retinitis pigmentosa (RP) and Leber congenital amaurosis (LCA) [6,10]. While both diseases cause primary rod cell death, they remain quite different from one another. LCA is often described as a more severe degeneration on the basis of age (early childhood) and rate of progression (blindness prior to or into early adulthood). Regarding genetic inheritance, LCA is usually recessively inherited. In contrast, RP maintains a more heterogenous group of diseases that includes dominant, recessive, mitochondrial, and X-linked inheritance, as well as a varying age of onset ranging from childhood to adulthood. Disease progression in RP is also varied from mutation to mutation and individual to individual [14,15]. LCA, a less common disease, affects 1:50,000 to 1:100,000 live births, while RP affects approximately 1:3000 to 1:5000 live births [6,10,14]. CRDs result in cone cell loss first, which, in humans, causes a rapid loss of visual acuity and central vision due to cone cell death in the foveal/macular region of the retina [1,5,11]. Diseases in this category include Stargardt’s disease and other macular dystrophies and are also able to be inherited heterogeneously, though less so compared to RP, and occur at a prevalence comparable to that of LCA [1,5,11,16]. Both RP and CRDs, unlike LCA, are capable of being inherited in a syndromic manner (i.e., the retinal disease maintains one phenotype in a primary, multisystemic disease such as Bardet–Biedl syndrome or Usher syndrome) or in a non-syndromic manner, with only the retina being affected [1,3,5,6,10,11,16,17,18,19]. Regardless of whether the IRD presents as a RCD or CRD, the ultimate outcome for the patient is blindness. The remainder of this review will focus on the underlying gene-dependent molecular mechanisms governing IRD pathogeneses, with our focus on the specific cellular organelles and molecular pathways affected by mutations in IRD-causing genes identified by perusing the literature, as well as RetNet (https://web.sph.uth.edu/RetNet/disease.htm, accessed on 28 August 2022). This review covers a majority of these genes, with an emphasis on those that are more commonly mutated.

## 2. Materials and Methods

### 2.1. Fluorescent Immunohistochemistry (fIHC)

Whole eyes from C57B/6J mice at 3 months of age were enucleated and fixed in 4% paraformaldehyde in PBS, pH 7.4 overnight at 4 °C. Fixation was quenched in 100 mM glycine in PBS, pH 7.4 for 10 min at room temperature and subsequently washed in PBS. Eyes were dehydrated with 30 min incubations in a graded ethanol series (50%, 70%, 85%, 95%, and 100%), then cleared via a 30 min incubation in graded xylenes (2:1, 1:1, and 1:2 ethanol:xylenes) and two 30 min incubations in 100% xylenes. Eyes were then infiltrated with paraffin using a graded paraffin series with 30 min incubations in 2:1, 1:1, and 1:2 xylenes:paraffin and two subsequent 1 h incubations in 100% paraffin. Paraffin-embedded tissue was then sectioned at 8 µm and sections deparaffinized and rehydrated, treated using heat-mediated antigen retrieval by heating slides at 95 °C in sodium citrate buffer (10 mM sodium citrate, 0.05% Tween-20, pH 6.0) for 1 h, washed in PBS twice, and subsequently blocked in 10% goat serum/5% BSA/0.5% TritonX-100 in PBS for 30 min at RT. Primary antibodies against markers for specific retinal cell types, cytoskeletal components, and IRD-associated genes were then applied at recommended dilutions and incubated overnight at 4 °C (Table 1). Slides were then washed in PBS, pH 7.4 three times for 10 min each. For standard fIHC, post-washing, slides were incubated in secondary antibodies conjugated to either AlexaFluor488l AlexaFluor568; or AlexaFluor647 (A21121, A21131, A21141, A11006, A21134, A11041, A11036, A21236, A21240, A21241, and A21449; ThermoFisher; 168 Third Avenue, Waltham, MA, USA) at 1:400 dilutions for 1 h and nuclei stained using a 1:10,000 dilution of 14.3 mM DAPI (D21490; ThermoFisher; 168 Third Avenue, Waltham, MA, USA) and mounted using Prolong Diamond Antifade mountant (P36961; ThermoFisher; 168 Third Avenue, Waltham, MA, USA). For multiplexed fIHC labeling, HRP-conjugated secondary antibodies were used (A16078, A10551, G21234, and A16005; ThermoFisher; 168 Third Avenue, Waltham, MA, USA), and, post-washing, tyramide signal amplification (TSA) was performed for 2 min with a 50 mM Tris-HCl, pH 7.4 reaction buffer containing 0.0015% H_2_O_2_ and tyramide reagent conjugated to either AZDye488; AZDye568; or AZDye647 (1538, 1541, and 1543; Click-ChemistryTools; 8341 E Gelding Drive, Scottsdale, AZ, USA 85260). After TSA, slides were washed in PBS three times and subsequently heated to 95 °C in PBS for 30 min to remove antibodies and subsequently probed using the second or third antibody in the multiplex label. After all TSA was performed, slides were stained with DAPI and washed in PBS, pH 7.4 four times and mounted using Prolong Diamond Antifade mountant (P36961; ThermoFisher; 168 Third Avenue, Waltham, MA, USA). After drying overnight, all sections were imaged using a Zeiss 710 laser scanning confocal microscope (LSM) using a 40× objective with 1.3 numerical aperture (NA) or 63× objective with 1.4 NA with or without an associated 1.7× zoom (~100×).

## 3. The Photoreceptor Outer Segment

The photoreceptor outer segment is a highly specialized primary cilium containing hundreds of membranous discs whose primary function is to house the components of the phototransduction cascade, which is responsible for converting light into a visual signal. A plethora of genes associated with IRDs affect some aspect of the outer segment, be it phototransduction, ciliary formation or trafficking, or disc morphogenesis, as seen detailed below.

### 3.1. Phototransduction

The phototransduction cascade is the pathway responsible for converting photons into an electrical signal that the brain is meant to interpret (Figure 2). Rods (and cones in bright light) are critical in this process, as they contain opsins, a group of specialized G-protein coupled receptors (GPCRs) that, when covalently bound to their ligand 11-*cis* retinal, are capable of absorbing visible light [20,21,22,23]. 11-*cis* retinal is one of the few GPCR ligands that covalently links to its receptor. In the case of retinal, the linkage is made through a protonated Schiff’s base [24,25,26]. When photons enter the eye and reach the retina, they are absorbed by the 11-*cis* retinal, converting it to all-*trans* retinal. Depending on the opsin to which it is bound, the wavelength of light the receptor absorbs changes, allowing for color discrimination [27,28]. Upon photoisomerization, activated opsins bind to and activate the heterotrimeric G-protein transducin (G_t_). Opsin-initiated activation of G_t_ results in the exchange of a GDP (inactive G_t_) for GTP (active G_t_) on the α subunit of G_t_ (G_tα_). G_tα_ dissociates from the remaining β and γ subunits and activates cyclic guanosine monophosphate (cGMP) phosphodiesterase 6 (PDE6). PDE6 is a highly efficient enzyme that hydrolyzes cGMP, a gate-keeping molecule that holds the cyclic nucleotide-gated (CNG) cation channel open into GMP. The decrease in intracellular cGMP concentration results in CNG channel closure and a subsequent hyperpolarization of the photoreceptor [20,21,22]. This causes glutamate release at the synapse to slow, resulting in signal propagation to the bipolar cells for eventual relay to the brain. Once turned on, the cascade must also be turned off. As described, phototransduction is a multistep process that includes numerous proteins (Figure 3). In the case of rod photoreceptors, the primary cells affected in RP, recoverin, an inhibitory protein of rhodopsin kinase (GRK1), releases its inhibition upon lower calcium levels subsequent to CNG channel closure. GRK1 then phosphorylates rhodopsin at its carboxyl terminus, a signal to the protein arrestin to cap the G_t_ activation site of rhodopsin [20,23,29]. With the site capped, G_t_ can no longer be activated. During rhodopsin shut off, G_t_ actively hydrolyzes GTP to GDP through an intrinsic GTPase activity, and this activity is further enhanced by a protein complex consisting of the regulator of G-protein signaling 9 (RGS9), RGS9 anchoring protein (R9AP), and guanine nucleotide-binding protein β5 (Gβ5) [30,31,32,33,34,35,36,37,38]. The decreased calcium levels serve another purpose of releasing the inhibition of GCAPs (guanylate cyclase activating proteins), which results in the activation of GCs (guanylate cyclases) of the retina that then restore the cGMP levels, effectively opening the CNG channels once again [9,11,12,21,39,40,41]. Ultimately, the all-*trans* retinal releases from the opsin molecule and is replaced with another molecule of the 11-*cis* retinal, which results in silencing the rhodopsin signaling and allows arrestin to disengage and the phosphates to be removed from the carboxyl terminus.

In the light activation pathway of phototransduction, there are a multitude of genes that, if mutated, have been shown to result in IRDs. *RHO* encodes one of the most critical proteins within the phototransduction cascade, rhodopsin (RHO), and has been identified as causing a considerable number of IRDs if mutated. Specifically, 30% of dominant RP cases have been linked to mutations in *RHO* [14,42,43]. RHO is a class A GPCR that, when activated by light, initiates the first step in vision. Failure of this protein to properly function causes RP by multiple mechanisms, including protein misfolding and aggregation (the most common mechanism of *RHO*-induced RP) and receptor mislocalization (the most severe forms of *RHO*-induced RP), as well as others such as constitutive activation and loss of G_t_ binding [14,43,44,45,46,47]. *OPN1LW*-*OPN1MW*, genes on the X chromosome, encode for red and green cone opsins, respectively. Mutations in these genes have been shown to present with X-linked cone dysfunction with and without degeneration [48]. *GNAT1* and *GNAT2,* the genes encoding the α subunit of G_t_ in rods and cones, respectively, are necessary for propagation of the photon-induced signaling cascade, which continues with the activation of PDE6 and closure of the CNG channels with subsequent hyperpolarization. Therefore, *GNAT1* and *GNAT2* mutations are causative for IRDs through a loss of phototransduction or a gain-of-function resulting in constitutive G_t_ signaling [49,50,51,52]. The next protein involved in phototransduction progression is PDE6, a key regulator in phototransduction functioning to hydrolyze intracellular cGMP, resulting in the closing of CNG channels and subsequently causing a hyperpolarization in the membrane potential and propagation of the visual signal. PDE6 is a heterotetrameric enzyme composed of an α, β, and two γ subunits, in which the α and β subunits constitute the catalytic core while γ functions as the inhibitory subunit of PDE6 in both rod and cone cells. Mutations in genes encoding these subunits have been associated with IRDs. For instance, mutations in rod PDE6 subunit genes *PDE6A*, *PDE6B*, and *PDE6G* (encoding the α, β, and γ subunits, respectively) have been shown to cause autosomal recessive RP. Since PDE6 is critical for phototransduction, the discussed mutations not only impede the cascade but lead to an elevated level of cGMP, which rapidly becomes toxic to the cell [16,53,54]. *PDE6C* is a gene that encodes for the α subunit of the cone photoreceptor PDE6. The inhibitory γ subunit of cone PDE6 is encoded by *PDE6H* and is also implicated in IRDs [55]. In addition, *AIPL1,* a gene encoding chaperone aryl hydrocarbon receptor (AhR)-interacting protein-like 1, is critical to the functional integrity of PDE6 by promoting its correct folding and assembly. Mutations in *AIPL1* have been associated with IRDs, including cone-rod dystrophies and LCA [56]. CNG channels, functioning directly downstream of PDE6, have also been linked to IRDs via mutations in genes encoding its α and β subunits that serve as the primary channel pore and regulatory calmodulin-binding sites, allowing for intracellular calcium level sensing and trafficking of the entire channel complex to the outer segment of the photoreceptor cell, respectively. Mutations in cone CNG channels, including *CNGB3* (encoding the β subunit) and *CNGA3* (encoding the α subunit), have been observed to cause autosomal recessive achromatopsia, a disorder characterized primarily by the absence of color vision [57,58]. On the other hand, mutations in the rod CNG channel’s α and β subunits, encoded by *CNGA1* and *CNGB1*, respectively, have been associated with RCDs [5,59]. Mutated β subunits of the CNG channels can also affect channel regulation through calcium binding, as well as disc formation issues associated with the amino terminal GARPs (glutamic acid rich proteins), which have been shown to associate with both PDE6 and the disc proteins peripherin-2 (PRPH2) and rod outer segment membrane protein-1 (ROM-1) (discussed subsequently in Section 3.3) [60,61,62,63,64].

Additionally, IRD-associated gene mutations have been identified in components involved in phototransduction deactivation. Arrestin-1 is a pivotal protein involved in phototransduction, specifically for RHO deactivation. It does so via binding activated RHO after phosphorylation by GRK1, blocking G_t_ molecules from binding RHO and preventing the subsequent steps of phototransduction from taking place. The binding of arrestin-1 to RHO allows for the timely return of the photoreceptor cell to its inhibited dark state. Mutations in *SAG*, the gene coding for arrestin-1, have been shown to cause RP by causing the rod cell’s dysfunction in returning to its dark state and, therefore, preventing further phototransduction, which can cause both IRD and congenital stationary night blindness (CSNB) [52,65,66,67,68]. Interestingly, there are no known mutations in GRK1 that cause IRD but have been shown to cause CSNB [69]. Other gene mutations, such as in *GNAT1* and *GNAT2*, have also been identified as factors affecting phototransduction deactivation. As previously discussed, *GNAT1* and *GNAT2* mutations have been shown to cause IRD through constitutive G_t_ signaling. However, G_t_ maintains intrinsic GTPase activity, which can also be hindered through mutations in *GNAT1* and *GNAT2*. With that, these mutations have been shown to cause a loss or slowing of phototransduction deactivation [70].

As previously discussed, the closure of CNG channels via the hydrolysis of intracellular cGMP to GMP is important in the progression of phototransduction. In relation to this mechanism, the replenishment of intracellular cGMP is essential to the restoration of open CNG channels in order to deactivate phototransduction and return to the dark state. Photoreceptor guanylate cyclase, GC-D, is a calcium-sensitive protein that functions to synthesize intracellular cGMP and is further activated and regulated by the calcium-binding guanylate cyclase activating proteins or GCAPs. Loss-of-function mutations in *GUCY2D*, encoding GC-D, eventually lead to low cGMP production from GTP and therefore prolong the activity of the phototransduction cascade, inducing degeneration [41]. *GUCA1A* and *GUCA1B* encode GCAP1 and GCAP2 that sense changes in cytoplasmic-free calcium, communicate it to GC-D, and therein activate it. Abnormalities in these genes have been implicated in RP through dysregulation and dysfunction of the GC-D protein, causing low cGMP production and cation imbalances [9,11,12,13,40,71].

The phototransduction cascade maintains many important channels and transporters for both dark- and light-activated signal propagation, as multiple components are involved in the regulation of membrane potential via the transport of ions. Similar to CNG channels, Na^+^/Ca^2+^-K^+^ exchanger 1 (NCKX1), a potassium-dependent sodium/calcium antiporter, functions to balance the intracellular calcium levels in rod outer segments. Mutations in *SLC24A1*, the gene encoding this antiporter, have been linked to causing autosomal recessive congenital stationary night blindness (CSNB) due to the partial or complete loss of ion transfer, resulting in abnormal levels of intracellular calcium concentrations [72,73]. Additionally seen to cause CSNB, specifically CSNB2A, is a dysfunction in the L-type voltage-gated calcium channel (VGCC), caused by a mutation in *CACNA1F*, the gene encoding the pore-forming α1F subunit of this channel. These channels, localized to photoreceptor terminals, function in mediating the inward calcium current required for the subsequent release of glutamate. The absence of this function via genetic mutation causes impaired or absent synaptic transmission from both rod and cone photoreceptor cells [74,75]. Finally, the activation range of potassium channels within cone and rod photoreceptors plays an important role in setting the photoreceptor resting potential. Kv8.2, a voltage-gated potassium channel subunit, acts as a voltage response modulator via shifting the activation range of potassium channels. *KCNV2*-associated retinopathy, an autosomal recessive cone-rod dystrophy, has been associated with gene variants in *KCNV2,* which codes for Kv8.2 and is currently under investigation as a potential target for gene therapy [76]. Interestingly, purine nucleotide synthesis is also important in maintaining the phototransduction cascade. Insoine-5′-monophosphate dehydrogenase, IMPDH (coded by *IMPDH1*), is the first committed and rate-limiting step enzyme in de novo purine biosynthesis, functioning to catalyze the conversion of inosine monophosphate (IMP) into xanthine monophosphate (XMP) that subsequently produces downstream GMP and GTP. IMPDH has been identified as essential in sustaining the intracellular GTP pool. Mutations in *IMPDH1*, coding for isoform IMPDH1 in retina, have been linked to causing IRDs such as autosomal dominant RP and, rarely, LCA. This is due to photoreceptor sensitivity to a required balance of purine nucleotides, including cGMP, which is a key signaling molecule in phototransduction [77,78,79]. In addition to *IMPDH1, PRPS1*, coding for phosphoribosylpyrophosphate synthetase 1, the enzyme catalyzing the first step in general nucleotide synthesis causes IRD, similar to *IMPDH1* [77,80,81,82].

### 3.2. Ciliary Formation/Trafficking

Cilia are small, projecting organelles that, until the last 20 to 30 years, were thought to be mostly vestigial. These organelles are now known to exist on almost all cell types and are essential to the cell’s survival, so that losing them can ultimately result in mortality [17,83,84]. Two types of cilia exist, motile and primary, also known as non-motile or sensory cilia. These ciliary types primarily differ in their structure and function. Motile cilia bear a 9 + 2 bundle of microtubules and, as their name suggests, are capable of movement and serve this function in regions such as the esophagus, lungs, and sperm (flagella), where they allow for cellular mobility (sperm) or the active movement of such things as mucus, food, etc. Primary cilia, however, have a 9 + 0 bundle of microtubules and serve a much different function dependent on cell type and tissue expression, from sensing urine flow in the kidneys to dictating the number of digits on a developing embryo’s hands [17]. In the retina, one of the primary roles of the cilium is established in the photoreceptor outer segment [85] (Figure 4). Within the photoreceptor outer segment, phototransduction proteins are integrated in, or associated with, the disc membranes (whose formation is discussed in Section 3.3) and the plasma membrane. The photoreceptor cilia, as sensory cilia, are arranged in a 9 + 0 bundle of microtubules coming from a single, tubulin-composed centrosomal basal body [86,87]. This cilium is termed the connecting cilium (CC), as it is a cellular and molecular highway for outer segment proteins (soluble proteins and integral membrane or membrane-associated proteins) and the vesicular transporters carrying them. One example of a major molecular transport function served by the CC is the flux of G_t_ into the inner segment and arrestin into the outer segment during phototransduction [88,89]. The movement between the inner segment and outer segment via the CC is driven by a process called intraflagellar transport (IFT) in which IFT proteins, kinesins, and dyneins transport a variety of products to their correct locations. Eventually, the tips of these cilia differentiate to form the outer segment, which contains RHO and other proteins necessary for phototransduction. Any abnormality in the generation of these cilia have been shown to lead to a lack of normal rod and/or cone function and diseases such as RP and LCA [90].

One of the major genes causing disease is *RPGR*. *RPGR* accounts for over 70% of X-linked RP and over 20% of non-syndromic RP [10,91,92]. *RPGR* codes for retinitis pigmentosa GTPase regulator (RPGR). This protein, when lost, results in the lack of any ciliary development and severe retinal diseases from birth. In its absence, the CC fails to form, and retinal cells begin to die from as early as 6 months of age in mice [92]. In a similar sense as *RPGR*, mutations in *RPGRIP1* also cause retinal degenerations. *RPGRIP1* encodes retinitis pigmentosa GTPase regulator interacting protein 1 (RPGRIP1), which plays an important role in the CC of photoreceptor cells. Similar to other ciliary proteins, losing functional RPGRIP1 results in the loss of proper vesicular protein trafficking and inner/outer segment flux, causing cell death [8,93]. Recent studies have identified biallelic missense mutations in RPGR-interacting domains of *RPGRIP1* to cause less severe forms of cone-rod dystrophies, while biallelic null mutations in this domain cause more severe forms [93]. *RP1L1* is a gene that encodes retinitis pigmentosa 1-like 1 (RP1L1), which encodes a component of the photoreceptor cilium and axoneme, the core structure of the cilium. Mutations in this gene cause photoreceptor disease and phenotypic retinal degeneration due to a loss of the synergistic interactions between it and RP1, causing a subsequent lack of ciliary formation [94,95]. In conjunction with *RP1L1* is *RP1*. *RP1,* coding for retinitis pigmentosa 1 (previously known as oxygen-regulated protein 1), is a photoreceptor-specific protein that localizes to the ciliary axoneme of outer segments, where it is thought to assist in horizontal disc orientation from the CC and, thus, ciliary, disc, and outer segment formation [90,95,96]. Due to this, mutations in *RP1* result in malformed cilia and outer segments. *FAM161A* is another IRD gene that encodes FAM161 centrosomal protein A, which is primarily localized to the CC, basal body, and adjacent centrioles. It has been shown as a part of microtubule organizing centers within the retina. The stabilization of the microtubules within the retina is critical to maintain the microtubule tracks in photoreceptor transport [97,98]. *TOPORS,* a gene encoding for a topoisomerase 1-binding arginine/serine rich (TOPORS), functions as a dual E3 ubiquitin and SUMO1 ligase and has been implicated in RP when mutated. This protein localizes to the basal bodies of the CC within photoreceptors and has been proposed to play a role in the regulation of primary cilia-dependent photoreceptor development and function. Within the cilium, TOPORS helps to localize and nucleate the microtubules from the centriole. Without this, cilia are incapable of properly developing, and retinal degeneration ensues [99,100,101,102].

The cilium also needs to be regulated in multiple aspects, including length, stability, and biomolecular composition. This is accomplished by multiple genes controlling protein trafficking, length, gating, etc. *CEP290* is a gene that encodes for a critical transition zone protein needed within the ciliary base. The transition zone functions as a diffusion barrier that, if mutated, perturbs the proper flux of inner/outer segment proteins, resulting in degeneration. CEP290 recruits DAZ-interacting zinc finger protein 1, which coordinates early ciliary membrane formation and transition zone assembly. Mutations in this gene lead to a lack of proper transition zone assembly and consequential retinal degenerations such as RP and LCA [103]. Basal exon skipping has been proposed as an underlying mechanism resulting in disease-causing CEP290 mutations [104]. *EYS* is also a gene that has been investigated and shown to have mutations that cause IRD. Its protein product, eyes shut homolog (EYS), has been shown in photoreceptor cells to maintain the ciliary axoneme in a stable formation. It preferentially localizes to the cytoplasm of the cilium and is used as an organization tool. EYS is essential for the structural maintenance of the ciliary pocket and organization of the CC of vertebrate photoreceptors. Mutations in *EYS* can cause disruption of the photoreceptor structure and subsequent CRD [102,105,106]. One ciliary regulator is *SPATA7*, which encodes for spermatogenesis-associated protein 7 (SPATA7). SPATA7 is localized to the CC and the RPE. Loss of this protein is shown to alter protein trafficking, including RHO and RPGRIP1 in the CC. Therefore, abnormalities in this protein cause retinal degeneration [107]. *IFT140* is a transport gene that is critical for proper retinal functioning. It encodes part of the IFT-A complex, which functions within IFT specifically by the retrograde transport of proteins from the ciliary tip. Biallelic variant mutations in this gene, including both missense and frameshift, have been seen to cause IRDs through abnormal protein localization and photoreceptor degeneration [19]. *LCA5* is also involved in the IFT of photoreceptor cilia, as deletions show impaired IFT [108]. The *MAK* gene codes for male germ cell-associated kinase (MAK), which regulates the length of the primary cilium in a variety of cell types. Within the retina, MAK deficiency results in retinal degeneration characterized and induced by the elongation of photoreceptor connecting cilia [109]. Oral-facial-digital syndrome 1 (OFD1) is another protein critical in regulating the photoreceptor cilium length and number. OFD1 additionally functions to provide a neuroprotective effect from oxidative stress and apoptosis. Insertions between exons 9 and 10, causing a frameshift in *OFD1* transcripts, the gene encoding OFD1, lead to the loss of primary cilia and therefore consequent photoreceptor degeneration [110,111]. RP2, encoded by *RP2,* is an outer segment trafficking protein and shares homology with tubulin-specific chaperone cofactor C. RP2 acts as a GTPase-activating protein (GAP) for ADP-ribosylation factor-like 3 (ARL3). ARL3, localized primarily to photoreceptor-connecting cilium, functions in trafficking lipidated proteins that are critical in phototransduction, such as G_t_, GRK1, and PDE6, to the outer segment. Abnormalities in ARL3 activation caused by RP2 dysfunction prevent the transport of these proteins and consequentially result in an RP phenotype [112,113]. *AHI1* is another gene which protein product is critical for ciliary trafficking mechanisms. It encodes for Abelson helper integration site-1 (AHI1), a signaling and scaffolding protein localized to photoreceptor cilia. Failure of this protein to function properly also leads to failure of the outer segment to properly assemble beyond the axoneme, leading to retinal dystrophy [114,115].

Though several genes associated with IRDs and the primary cilia are non-syndromic, a multitude of the genes in this category are syndromic, and several of the genes overlap between syndromic and non-syndromic (Figure 5). Interestingly, different types of mutations in these genes are a major contributing factor to the disease manifestations, be it non-syndromic or syndromic. As an example, a nonsense mutation in a gene associated with cilium development yielding no functional protein would be more likely to cause syndromic disease, while a nonsynonymous missense mutation in the same gene may cause a loss of a retina-specific interaction (i.e., LCA/RP proteins such as CEP290), causing non-syndromic IRD. Below details this subset of IRD genes.

The BBSome is a complex critical for ciliogenesis and ciliary protein trafficking through the recruitment of Rabin8 and Rab8 [116,117,118]. It is encoded by several genes that are implicated in a syndromic retinal dystrophy called Bardet–Biedl syndrome. These include *BBS1-22* [119]. While 22 different genes are associated with Bardet–Biedl syndrome, many serve redundant functions and will not be discussed here. *BBS1* has been shown to affect retrograde trafficking of GPCRs out of primary cilia, and mutations in the gene have been seen to cause retinal degeneration [120,121]. Mutations in *BBS2*, *BBS4*, *BBS5*, *BBS7*, *BBS9*, *BBS17*, and *BBS18* all result in an RP phenotype due to a loss of ciliary formation and/or mislocalization of photoreceptor proteins, including RHO and arrestin-1 [116,119,120,121,122,123,124,125,126,127,128,129,130]. *BBS8* (*TTC8*) is a gene that has been associated with Bardet–Biedl syndrome and non-syndromic RP that codes for a BBSome member and is necessary for the proper trafficking of ciliary/outer segment proteins. The functional loss of BBS8 results in the mislocalization of outer segment proteins to the inner segment and axons, as well as cytosolic and axonal protein mislocalization to the outer segment and other cellular domains. The functional loss of BBS8 also results in the loss of ciliary formation [18,131]. BBS10 is a part of the BBSome complex assembly process and functions to regulate the interaction of the BBS6–BBS12–BBS7 intermediate step of BBSome assembly [132]. Another gene with mutations exhibiting a Bardet–Biedl syndrome phenotype is *MKKS/BBS6*. The association of MKKS and CEP290 (also known as BBS6 and BBS14, respectively) at the ciliary transition zone and basal body affects the integrity of multiprotein complexes, including the BBSome. These are essential functions within ciliary biogenesis, and mutations in the gene show a lack of ciliary function and the associated IRD [133,134]. Another of the BBS proteins that is implicated in ciliary dysfunction is BBS3. *BBS3* encodes for Arf-like 6 GTPase (ARL6), which recruits the BBSome to the ciliary membrane to interact with its trafficked proteins. Therefore, without proper function, the BBSome does not localize to the correct location, and the same mechanism of disease follows [135]. *SDCCAG8*, also known as *BBS16,* causes Bardet–Biedl syndrome through the loss of ciliary formation due to the carboxy terminus of the protein product, SDCCAG8, being directly involved in the process [136,137,138,139]. The final BBS-associated gene discussed that causes Bardet–Biedl syndrome is *IFT27* (also known as *BBS19*). Intraflagellar transport 27, coded by *IFT27,* is crucial for BBSome trafficking and Sonic hedgehog signaling. Cells depleted of these proteins due to an *IFT27* mutation accumulate abnormal collections of IFT complex B proteins and cause phenotypic Bardet–Biedl syndrome [140]. *IFT172* and *IFT74* also code for IFT proteins with similar effects, while *BBS21* codes for cilia- and flagella-associated protein 418 (CFAP418), which is found at the base of the connecting cilium and associates with other proteins involved in the transition zone and disc formation (discussed in Section 3.3) [141,142,143,144].

Other genes causing IRD through a syndromic form have been linked to Usher syndrome, which results in both RP and typically deafness and presents as one of three types (USH1, 2, or 3). *MYO7A* (or *USH1B*) is a known Usher-associated gene that encodes for unconventional myosin VIIa. This protein is localized in the apical region of the RPE and the ciliary membrane of the photoreceptors. Its function lies in the apical localization of RPE melanosomes and in the removal of phagosomes from the apical RPE for their eventual delivery to the basal RPE lysosomes. When *MYO7A* is mutated, opsin accumulates abnormally in the transition zone of the cilium, which causes retinal dysfunction, as well as a deficiency in autophagy for RPE cells [145,146,147,148,149]. *USH1C* is a gene needed for proper photoreceptor ciliary and synaptic development that encodes for the PDZ-domain-containing protein harmonin. Harmonin functions as a key scaffolding protein that other USH proteins bind. This is essential for the synaptic and ciliary maturation and maintenance of Müller glial cells. Without proper cilia and synaptic development and glial maintenance, retinal cells degenerate [150]. Cadherin-related 23 (CDH23 or USH1D), encoded by *CDH23,* is a member of the cadherin family that is comprised of calcium dependent cell-cell adhesion glycoproteins and are commonly known for their role in maintaining and establishing the organizational integrity of stereocilia within the hair cells of cochlea [151]. The role of CDH23 is well characterized in the development of hair cells of the inner ear but has yet to be further elucidated in its retinal function. Mutations in the transmembrane cadherin-like domain region of *CDH23* have been linked to phenotypes presenting in Usher syndrome [151]. Protocadherin-15 (PCDH15 or USH1F), encoded by *PCDH15*, is localizes to the photoreceptors where it participates in outer segment scaffolding and mutations cause loss of this structural component and degeneration [152,153,154,155]. Through the same mechanism, *USH1G* encodes scaffold protein containing ankyrin repeats and SAM domain (SANS). SANS is known to interact with myomegalin and other USH proteins and is needed for organizing the network of USH proteins necessary for the mechanical stabilization of the outer segment, as well as the intracellular transport processes. Therefore, this lack of USH proteins consequently leads to additional retinal degeneration [156,157,158]. *USH2A*, coding for USH2A or Usherin, is one of the most common genes associated with Usher syndrome. The function of USH2A is to interact with ADGRV1 and bridge the gap between the membranes of photoreceptor CC and the periciliary region. Lack of this gene leads to the clinical manifestation of Usher syndrome due to the lack of the described interaction [154,159]. *ADGRV1*, also known as *USH2C,* is a gene that encodes for adhesion G protein-coupled receptor V1 (ADGRV1). It is located in the periciliary region between the inner and outer segments and thought to function in stabilizing the CC. Defects in this gene have been shown to result in abnormalities in the fibrous links and adhesion between sensory cells that can lead to photoreceptor dysfunction [160,161]. *USH2D* codes from the protein whirlin, a gene named for the effect it has in mice bearing mutations in the gene where a loss of vestibular function causes the mice to circle. This gene also results in retinal degeneration as it interacts with SANS, USH2A, and ADGRV1 [157,162,163,164]. *USH3A* is a gene encoding clarin-1, a four transmembrane-domain protein whose role lies in the excitatory ribbon synapse junction in hair cells of the ear, as well as in analogous synapses within the retina. It has also been localized to Müller cells and photoreceptor inner segments where it associates with microtubules, the framework components of the primary cilium [165,166,167,168,169]. Clarin-1 may also have a role in cell-cell adhesion between Müller cells and photoreceptors. A pathophysiology shared between the vestibular and visual systems presents itself when clarin-1 is incapable of proper intracellular trafficking and plasma membrane insertion, causing deafness and RP in Usher syndrome patients and have been linked to deletions in both exon 2 and 3 of *USH3A* [166].

Aside from Bardet-Biedl Syndrome and Usher syndrome, other systemic disorders are associated with ciliary defects (called ciliopathies) and several of the genes overlap between disorders. Alström syndrome (ALMS) is a vision disorder primarily characterized by CRD and hearing loss and is attributed to a mutation in *ALMS1* gene, encoding ALMS1 [170,171]. Although the protein’s direct function remains unclear, ALMS1 was shown to have a role in both endosomal and ciliary transport. It localizes to the centrosomes and basal bodies of ciliated cells where it helps with trafficking of transferrin, GLUT4, Notch1 and others [172]. Therefore, without proper functioning, protein trafficking becomes abnormal and initiates IRD [170,171]. *NPHP1* is another ciliary transition zone gene that encodes nephrocystin-1 (NPHP1). Its function is required to prevent infiltration of inner segment plasma membrane proteins during the photoreceptors outer segment development. It also has been shown to have a function in protein trafficking within the ciliary compartment. Therefore, proteins become mislocalized with mutations and photoreceptor dysfunction occurs [173]. *NPHP4* is a ciliary gene encoding for nephrocystin-4 (NPHP4) that when mutated, patient photoreceptor outer segments fail to form properly and certain proteins mislocalize to the inner segments and outer nuclear layer. This has suggested a function for NPHP4 in intraflagellar transport or sensory signal transduction [174]. *NPHP5* mutations are also implicated in retinal dystrophies, as it encodes a protein that functions to help localize parts of the BBSome complex (BBS2 and BBS5). Patients deficient in this gene have been shown to have mislocalizations of BBSome components and are unable to form connecting cilia and proper outer segments [175]. This group of *NPHP* genes collectively cause a disease called Senior-Løken syndrome. *CEP41,* coding for centrosomal protein 41, is necessary for regulation of tubulin glutamylation which is subsequently required for cilia disassembly and endothelial cell dynamics such as migration and tubulogenesis. Therefore, without this protein, cilia do not function properly and in turn cause retinal degeneration [176]. INPP5E or pharbin is a ubiquitously expressed phosphatidylinositol polyphosphate 5-phosphatase that is located within the primary cilia and functions to regulate the membrane composition of phosphoinositide. Mutations in *INPP5E* cause ciliary dysfunction and loss of ciliary localization of signaling proteins [177]. Working along with *INPP5E* is another gene, *MKS1* (also known as *BBS13*). *MKS1* encodes for Meckel syndrome 1 (MKS1) which is localized to the transition zone at the base of the cilium and functions to regulate ciliary INPP5E content via ADP-ribosylation factor-like GTPase-13B (ARL13B). Complete loss of the protein product of *MKS1* has been shown to negatively affect cilium formation through abnormal basal body docking [178]. Retinitis pigmentosa GTPase regulator interacting protein 1-like (RPGRIP1L) is a protein coded for by the *RPGRIP1L* gene which, by its name, is a protein with high homology to RPGRIP1 and interacts with similar protein partners. As one would expect, mutations in this gene result in a loss of proper ciliary formation in the retina and other organs causing the symptoms of a ciliopathy, including RP [179,180,181]. Intraflagellar transport complex A component IFT144, encoded by *WDR19,* is another ciliary protein functioning specifically in retrograde ciliary transport of cargo in the cilium. Abnormalities in *WDR19* expression, specifically a missense mutation, have been shown to express abnormal ciliopathies, including Sensenbrenner, Jeune Syndrome, and some forms of nephronophthisis [182]. Moreover, another gene indicated in IRDs is *KIF11,* coding for kinesin family member 11 (KIF11), a molecular motor protein known for its role in bipolar spindle formation and protein trafficking and primarily localized to the ciliary regions of the retina [183,184,185]. Mutations in *KIF11* have been shown to cause retinal hypovascularization due to stunted growth of retinal vasculature. The pathogenesis of mutant KIF11 is still under investigation [184,186]. Many of these genes are associated with other syndromic diseases such as Joubert syndrome and Meckel syndrome, both forms of ciliopathies. Mutations in the *CHM* gene, resulting in a loss of function for its product protein, Rab escort protein-1 (REP-1), has been linked to causing choroideremia (CHM) [187]. CHM is characterized as an X-linked disease stemming from a gradual degeneration of RPE, choroid, and photoreceptor cells resulting in peripheral field vision loss once reaching adulthood. REP-1 normally functions as a post-translational lipid modifier of vesicular trafficking and phagosome fusion protein, Rab GTPase [188]. Mutant CHM expression pathogenesis is currently under investigation in that researchers are currently attempting to create a stable knock-out mouse model [187,189]. Interestingly, some mutated genes, such as *SCAPER* and *ABHD12,* result in phenotypes associated with a ciliopathy, but no immediate biomolecular evidence has been presented as to how they cause these phenotypes. *SCAPER*, a gene identified in a group of patients observed to have multiple BBS phenotypes, have no directly observed function for SCAPER protein for the phenotype [190,191]. Another gene similar to *SCAPER* is *ABHD12*. This gene’s protein product is thought to be involved in cell signaling using the endocannabinoid system and lysophosphatidylserine [192]; however, its phenotype is a disease PHARC, which clinically appears comparable to Usher syndrome [192,193].

### 3.3. Disc Morphogenesis

Discs are a significantly critical component of the phototransduction cascade as they are the foundation and attachment point for the receptors, second messengers and enzymes necessary to run the process. These discs are located in the outer segments of photoreceptor cells and lack all other organelles. Instead, they contain a great number of discs that are large flattened membranous vesicles. Cone and rod photoreceptors differ in their disc morphologies. Rod photoreceptors possess “closed” discs separated from the plasma membrane while cones possess “open” discs formed by an invagination of the plasma membrane [194]. In addition, rod discs are held suspended separately from one another and held in place at the edges where they connect to the plasma membranes through protein-protein interactions [194]. As the formation of these discs within the rod photoreceptor is still a topic of ongoing research, they are believed to be formed by the evagination of the plasma membrane with the rods and eventually pinching off to form their own separate membrane bound structure [194,195]. Due to the requirement of these discs and their phototransduction contents, any dysmorphogenesis with these discs is implicative of forming retinal based diseases such as RP or LCA [196]. Several genes have been associated with disc formation, as well as their resultant diseases caused by mutation.

The primary gene associated with disc morphogenesis is *PRPH2* which codes for the PRPH2, a photoreceptor specific peripherin generally responsible for the flattening of the vesicles forming the discs and their attachment to the plasma membrane [195,197,198,199,200]. PRPH2 is restricted to the disc lamellae rim where it co-localizes with ROM-1, rod outer segment membrane protein 1. If *PRPH2* is mutated, discs fail to properly flatten and lose contact with the membrane [200]. Interestingly, ROM-1 allows for PRPH2 to be trafficked to the disc lamellae rim and their interaction is necessary for proper disc flattening. Without properly functioning ROM-1, PRPH2 would be trafficked elsewhere, and discs would fail to form. Thus, mutations in *ROM-1* cause IRDs similarly to loss of PRPH2 [90]. Another of these critical disc genes is *FSCN2*. In mice, *FCSN2* has been shown to code for the protein fascin actin-bundling protein 2 that functions to induce actin binding and bundling, a process critical in the morphogenesis of discs within the outer segment. Mutations in this gene in mice have showcased steadily worsening rod function over time in a manner that indicates FCSN2 as being critical for the elongation and maintenance of discs within the outer segment [201,202]. *PRCD* is also a gene necessary for disk formation. Mislocalization of the encoded protein leads to rapid degradation of the retina. In normal photoreceptor cells, discs are aligned in a precise way, but in *PRCD* mutated mice, these discs are misaligned and have bulges with an associated accumulation of vesicles in the outer segment. This leads to eventual toxicity and degeneration of the retinal photoreceptors [203]. *PROM1* codes for prominin-1, a transmembrane protein localized to the photoreceptor outer segment that is critical for assembly of discs, specifically the evagination and remodeling processes. Although not observed often in the population, its mutation has been shown to cause less rapid retinal degeneration than other forms, similar to that of mutations in *PRPH2* [204]. Mutations in *PCDH21,* which codes for protocadherin-21 (PCDH21), has been associated with IRDs. Its function is thought to be related to its location between the inner and outer segments. PCDH21 may be responsible for proper disc alignment, evagination, and tethering, making its loss through mutation detrimental to outer segment function and formation [90,205]. The final two genes to mention regarding disc morphogenesis are *CFAP418* (mentioned previously) and *PCARE.* These two genes are hypothesized to be crucial for disc formation and protein trafficking to the outer segment and the actin dynamic organization in the outer segment base, respectively, and both result in RP when mutated [142,206]. It’s still not currently known whether these genes are more associated with disc formation or ciliary formation. Regardless, the loss of proper outer segments and protein mislocalization suggest it could be either or both.

## 4. The Retinal Pigment Epithelium (RPE)

The RPE is a monolayer of cells located at the back of the eye between the retinal photoreceptors and their primary blood supply, the choriocapillaris. The RPE serves a multitude of functions for the retina, including producing the opsin chromophore 11-*cis* retinaldehyde, phagocytosing overgrown outer segments, excretion of metabolic waste to the bloodstream, absorbing stray photons, and assisting with general homeostasis. Many genes in the RPE are associated with IRDs when mutated and these are outlined below.

### 4.1. The Visual Cycle

Phototransduction, as covered in Section 2.1, relies on the replenishment of the chromophore 11-*cis* retinal, the inverse agonist of opsins. The visual cycle, or retinol cycle, can be summarized by a series of enzymatic reactions that result in the re-isomerization of all-*trans* retinal to 11-*cis* retinal [207] (Figure 6). In general terms, during phototransduction, when the all-*trans* retinal is cleaved from rhodopsin (or cone opsins) via hydrolysis, it is released from the opsin apoprotein and diffuses laterally through the photoreceptor disc membrane and is trafficked to the RPE using a series of enzymes and transporters [208]. To do so, the all-*trans* retinal reacts with the primary amine of endogenous phosphatidylethanolamine (PE) to form *N*-retinylidene-phosphatidylethanolamine (*N*-Ret-PE), the primary substrate for retinal specific ATP binding cassette transporter 4 (ABCA4). ABCA4, located in the outer segments of photoreceptor cells on the rim of disc membranes, functions as a flippase. The binding of *N*-Ret-PE to ABCA4 results in a confirmational change in which *N*-Ret-PE is released from its binding site towards the cytoplasmic leaflet of the disc membrane and is released from the PE as the all-*trans* retinal form [209]. Once in the cytoplasmic leaflet of disc membranes, the aldehyde of the all-*trans* retinal is reduced by all-*trans* retinol dehydrogenase (atRDH) with use of the cofactor NADPH, resulting in the production of all-*trans* retinol. Upon this reduction, the retinol is trafficked out of the photoreceptor outer segments using interphotoreceptor retinol binding protein (IRBP) to the RPE [210]. IRBP is a soluble lipoglycoprotein that preferentially binds all-*trans* or 11-*cis* retinoids and functions to transport retinoids between photoreceptor outer segments and apical RPE [211]. Once in the RPE, lecithin retinol acyl transferase (LRAT) catalyzes the transfer of a fatty acyl group from endogenous phosphatidylcholine to the all-*trans* retinol forming all-*trans* retinyl esters, formulating a nontoxic storage form of all-*trans* retinol [212]. A majority of retinyl esters are stored in hepatic stellate cells via the use of retinal binding protein 4 (RBP4) [213]. Regarding the storage of retinoids, the hydrophobicity of retinoids limits their ability to diffuse in aqueous environments, creating a barrier for transport to storage sites. RBP4, a member of the lipocalin family of proteins, functions to reversibly bind and sequester retinoids in order to facilitate their diffusion, with its activity significantly reliant on retinoid homeostasis. This activity is highest for the mobilization of retinoids to the liver in which retinyl esters are hydrolyzed to retinol and bind RBP4 to be carried through the bloodstream [214].

As all-*trans* retinyl esters are being formed along the smooth endoplasmic reticulum of RPE cells by LRAT; the *trans* isoform is then isomerized via retinoid isomerohydrolase retinal pigment epithelium 65 kDa (RPE65). RPE65 is primarily expressed in RPE cells and isomerizes retinyl esters to result in production of 11-*cis* retinol via a *trans*-to-*cis* alkene bond isomerization followed by a unique *O*-alkyl bond cleavage resulting in the dissociation of the ester group. The production of 11-*cis* retinol is the rate limiting step of the visual cycle [215]. After isomerization, 11-*cis* retinol dehydrogenase (11-cRDH) then oxidizes the alcohol groups of 11-*cis* retinol to an aldehyde, resulting in the formation of 11-*cis* retinal [210]. 11-*cis* retinal is then chaperoned by cellular retinaldehyde binding protein (CRALBP), which functions to facilitate intracellular transport of retinoids in the RPE to protect them from further esterification or thermal-isomerization [210,216]. As its function has been previously described, IRBP preferentially binds 11-cis retinal and traffics it back to the outer segment of photoreceptor discs. Now, 11-*cis* retinal inside of the photoreceptor discs can once again bind opsin to regenerate functional rhodopsin (or cone opsins) for phototransduction to continue [208]. This series of enzymatic reactions and trafficking of retinoids describes the dark adaptation regeneration of 11-*cis* retinal and is thought to be the primary mechanism of chromophore regeneration [208]. A secondary pathway, commonly referred to as the RGR (retinal G-protein coupled receptor) pathway, provides a direct photoisomerization of retinoids back into the 11-*cis* isomer. This can occur either in the RPE or the Müller cells, as RGR is known to be expressed in both cell types [217]. Photoisomerization in the RPE begins with chaperoning all-*trans* retinol via CRALBP. Under light conditions, RGR in complex with 11-cRDH, functions to directly photoisomerize all-trans retinol, resulting in the production of 11-*cis* retinal. Similar to the RPE65-based 11-*cis* retinal regeneration, CRALBP protects 11-*cis* retinoids produced by RGR from further esterification or photoisomerization effects [218].

*ABCA4*, coding for a flippase and member of the ATP-binding cassette (ABC) superfamily, is localized to the edge of disc membranes in photoreceptor outer segments and primarily functions as a retinoid transporter, as well as the RPE [219,220]. It has been shown that ABCA4 is highly expressed in the retina and functions in the transport of *N*-Ret-PE from the lumen of disc membranes to the cytoplasmic leaflet powered via ATP hydrolysis and hypothesized in the RPE to recycle the all-*trans* retinal present in phagocytosed discs [220,221]. In general, this prevents the accumulation of retinals and their toxic byproducts in disc membranes and the RPE. As previously described *N*-Ret-PE, the primary substrate of ABCA4, is synthesized via the reaction between all-*trans* retinal and endogenous phosphatidylethanolamine (PE), which results in the formation of pyridinium bis-retinoid (A2E), a prevalent lipid precursor to lipofuscin [222,223]. It has been shown that failures in ABCA4 function exacerbate the formation of A2E. The accumulation of cytotoxic biproducts such as lipofuscin are phenotypic hallmarks of multiple IRDs, including autosomal recessive Stargardt’s disease [222,224].

*RPE65* is critical to the visual cycle in which its protein product RPE65 functions to convert all-*trans* retinyl esters to 11-*cis* retinol. RPE65 catalyzes this the enzymatic isomerization of all-*trans* retinyl esters via a radical cation mechanism to yield 11-*cis* retinol that is subsequently oxidized to 11-*cis* retinal [225]. Without properly functioning RPE65, there is substantial reduction in the levels of 11-*cis* retinal in addition to the accumulation of retinyl esters in the RPE [226]. RPE65 is one of the first RPE-specific proteins that has been identified to be associated with human IRDs, most severely seen in LCA and mildly in autosomal recessive RP, diseases that can be generalized as blindness from birth or early childhood [227]. Interestingly, patients with RPE65 deficient LCA showcase preserved retinal structure prior to the decline of visual function, providing the opportunity for investigation of potential therapeutic intervention prior to permanent vision loss. There are currently several human clinical trials, including gene replacement and augmentation therapy options for RPE65-associated dystrophies underway, as well as one that has been FDA approved [228,229,230,231].

As previously described, the secondary pathway of the visual cycle that functions to provide direct stereospecific photoisomerization of retinoids into the 11-*cis* isomer, requires the activity of the gene product of *RGR*, a seven transmembrane domain receptor localized to the intracellular membranes of both RPE and Müller cells. RGR, a close relative to RHO, is coupled to all-*trans* retinal that is then, upon light exposure, isomerized to 11-*cis* retinal [232]. Through studies involving *RGR* knockout mice, it has been shown that RGR is necessary in maintaining a normal rate of 11-*cis* retinal production in both dark and light conditions [233]. However, specified ocular diseases have been rarely reported to be in direct association with an *RGR* mutation. Although, both heterozygous frameshift and missense human RGR mutations have been closely associated with autosomal recessive RP [234].

*LRAT*, coding for a member of the NlpC/P60 thiol peptidase protein family, plays a role controlling the concentration of retinoic acid in the RPE. LRAT functions in the visual cycle as the main enzyme responsible for the conversion of all-*trans* retinols into all-*trans* retinyl esters [235,236]. LRAT has additionally been identified in the regulation of intestinal retinoid biosynthesis via negative feedback regulation [237]. LRAT activity, which can be defined by the transfer of an acyl moiety from the -sn1 position of phosphatidylcholine to all-*trans* retinols, has been identified to play a key role in maintaining optimal retinoid levels throughout the body [238]. The first reported *LRAT* human mutation has been associated with an early onset, severe retinal dystrophies, including rod cone dystrophy and has been compared to phenotypes similar to that resulting from mutation in *RPE65*. Studies in *Lrat* ^−/−^ mice have showcased LRAT as a critical enzyme for photoreceptor health and ocular retinoid homeostasis due to its role in the uptake and retention of vitamin A in the RPE which greatly impacts the production of 11-*cis* retinal [239]. This phenomenon has been shown to attribute to pathology seen in LCA patients, which was observed to result in disordered vectorial transport of cone visual pigments lacking bound chromophore. Currently, *LRAT* gene replacement therapy is under investigation as a potential treatment for RP associated with *LRAT* loss of function mutation [240].

The *RDH5* gene, encoding for 11-*cis* retinol dehydrogenase, is abundantly expressed in RPE cells and functions in the oxidation of 11-*cis* retinol to 11-*cis* retinal prior to its transport to photoreceptor outer segments [241]. RDH5, in association with RDH11, plays an important role in avoiding the accumulation of 11-*cis* retinol in RPE cells, as well as providing photoreceptors with 11-*cis* retinal for proper visual cycle processes [242]. Mutations in *RDH5* have been found to result in an autosomal recessive pattern of fundus albipunctatus, also referred to as retinol dehydrogenase 5 retinopathy, which is a form of CSNB and complaints of delayed dark adaptation after bright light exposure attributed to a lack of 11-*cis* retinal. An oral 9-*cis*-retinoid therapy has been proposed to function in the recovery of retinoids via bypassing the function of RDH5 and is currently under clinical investigation [243]. *RDH12*, as mentioned, is a member of a dehydrogenase/reductase family and is highly expressed in photoreceptor cells outer segments. In association with RDH8, RDH12 converts all-*trans* retinal to all-*trans* retinol in order to prevent the accumulation of toxic aldehyde by-products during the visual cycle via the reduction of all-*trans* retinal [244,245]. The accumulation of aldehydes in photoreceptor outer segments is closely related to mutations in *RDH12* seen in ocular diseases, including up to 4% of the recessive form of LCA [246], as well as autosomal recessive retinitis pigmentosa, CRDs, and even macular dystrophies. *RDH12* mutations maintain a broad retinal phenotypic spectrum with significant age variability [247]. In studies involving mouse models with disrupted *RDH12* gene expression, although there is not a significant lack of chromophore regeneration, the retina was observed to become more susceptible to the toxic effects of light. With that, it is important to note a fundamental difference in the nocturnal nature of rodents in that their visual cycle processes retinoids slower than humans [248]. While the human retina’s visual cycle is much faster and the retina is twice as rich in cone pigments, it is susceptible to a more sensitive effect of toxic light damage. It has been proposed that patients maintaining an *RDH12* null allele may reduce the rate of retinal degeneration via avoidance and protection from intense illumination [248,249].

*RLBP1* encodes the visual cycle protein CRALBP, which functions to enzymatically facilitate the transport of retinoids, specifically 11-*cis* retinal, and protect it from further esterification/photo-isomerization and is expressed in abundance in both RPE and Müller cells [250]. CRALBP functions in the visual cycle by the reduction of spontaneous isomerization which allows for the ensured efficient yielding of 11-*cis* retinol and accumulation of aldehyde toxicity [251]. Mutations in *RLBP1* cause autosomal recessive retinal diseases such as retinitis punctata albescens, recessive RP, and Newfoundland RCD and is phenotypically characterized by the degree of delayed dark adaptation and macular atrophy [252,253,254]. As previously defined, RBP4 functions in the storage of retinyl esters in hepatic stellate cells via facilitated diffusion and is the only known specific binding protein in circulation, a process essential for retinoid homeostasis [214]. In previous studies, *RBP4* knockout mice resulted in the decreased uptake of retinol into the retina which subsequently caused an expected reduction in levels of chromophore, as well as impaired vision [255]. Interestingly, recent studies have shown that overexpression of *RBP4* in mice resulted in the early onset of retinal degeneration [256]. There have been studies identifying a homozygous splice variant in *RBP4* that has been associated with severe autosomal recessive RP in a pedigree of European ancestry in addition to a few members suffering from chorioretinal and iris colobomas [257]. *RBP4* mutations, however, are not limited to ocular diseases and contribute to a multitude of lipid homeostasis diseases, glucose and cardiovascular dysfunctions, and many other human conditions [258,259].

### 4.2. Phagocytosis

Another critical maintenance process within the retina is the turnover of the outer segment. This turnover of the outer segment region of photoreceptor cells involves the RPE, a region of single layered cells directly adjacent to rod outer segments [260]. As previously mentioned, these outer segments are turned over at rapid rates when exposed to light. Each day, around 10% of the outer segment is phagocytosed to allow for new disc formation at the outer segment base [260]. Specifically, respective light and dark associated molecules dopamine and melatonin, along with the circadian clock genes *PER1* and *PER2* and L-type Ca^2+^, K^+^ and Cl^−^ channels, have been shown to have a role in the rhythmic shedding of outer segments [261,262]. This leads to the outer segment being turned over completely around every 10 days. If this turnover is not properly regulated, it can lead to photoreceptor damage and/or a variety of visual dysfunctions. Of the many key functions done by the RPE, turnover of photoreceptor outer segments is one of the most critical when related to retinal dystrophies. This process repeats daily, so any malfunction in the required proteins necessary lead to abnormal balances and apoptosis [263].

One of the most critical of phagocytosis genes is *MERTK*. *MERTK* encodes a receptor, c-mer tyrosine kinase (MERTK), on the apical surface of the RPE that functions to internalize the outer segment prior to phagocytosis. As it is necessary for both MERTK to be present for phagocytosis and the retina needing to consistently phagocytose older portions of the outer segment, the mutations in the coding gene can cause buildup of shed photoreceptor disc membranes and the biomolecules contained within which can be significantly toxic to the retinal environment. Mutations in *MERTK* have been shown to cause steady retinal degeneration, leading to early signs of vision loss and blindness [264,265]. In addition, the IRD-associated protein ceramide kinase-like protein (CERKL), has been shown to regulate MERTK and thus influence the natural phagocytic turnover of the outer segment [266,267]. Therefore, abnormalities in CERKL functionality can induce either too much or too little turnover of the outer segment which requires a very tightly regulated balance [267].

A multifunctional gene, *TULP1*, has also been seen to cause IRDs. This gene codes for a protein, tubby-like protein 1 (TULP1), that localizes to multiple regions of the photoreceptor, ranging from the ribbon synapse up to the outer segments in which it functions in vesicular trafficking, endocytosis, and molecular recognition. Studies in mice maintaining deleterious mutations within *TULP1* have shown shortened outer segments and lower rod length early in development. This is quickly followed by apoptosis and complete loss of functional retinal tissue [268]. It has also been shown that TULP1 may also act as a recognition protein for MERTK to initiate outer segment phagocytosis by the RPE [269]. The gene *RAB28*, coding for a ubiquitously expressed, farnesylated small GTPase, has also been implicated in the proper phagocytosis of photoreceptor outer segments, specifically those of cone photoreceptors [270]. The protein-protein interactions between RAB28 and Kir7.1, an inwardly rectifying potassium channel, in the RPE microvilli is required for proper shedding of cone photoreceptor outer segments [270,271]. Nonsense mutations in *RAB28* have been identified to cause recessive cone-rod dystrophy 18, as well as RPE atrophy and progressive loss of visual acuity [270,271].

### 4.3. Other RPE Functions: Pigment Formation, Ionic Regulation, and Lipid Homeostasis

Aside from retinal/retinol recycling and phagocytosis, the RPE also plays other important roles in maintaining a retinal environment conducive for visual processes. One of these roles includes the protection of the retina from excessive illumination and UV light through the production of pigment. As seen in ocular albinism, the lack of retinal pigment (melanin) can lead to serious retinal degeneration attributed to lacking protection from light damage [272,273]. While pigment is important for light protection, it also plays a role in promoting expression of neurotrophic factors from the RPE. The gene *GPR143* plays a major role in this facet. It is hypothesized that the protein product G protein-coupled receptor 143 (GPR143) acts as a receptor for the reaction by-product of melanin synthesis, L-DOPA, and that by binding L-DOPA, GPR143 signaling causes release of pigment epithelium derived neurotrophic factor (PEDF), a pro-survival factor for the retinal neurons [274]. It has also been shown that concomitant mutations in both *GPR143* and cone opsin genes can result in foveal hypoplasia with monochromatism, as well as albinism type I [275].

In addition to pigment signaling, the RPE also works to regulate ionic homeostasis by using ion channels. Bestrophin-1 (BEST-1, coded for by the *BEST-1* gene) is a pentameric ion channel that localizes to the basolateral side of the RPE near Bruch’s membrane and at the endoplasmic reticulum [276]. Due to its localization and interactions with calcium channels, BEST-1 is thought to be necessary for proper maintenance of intracellular calcium stores and retaining proper RPE cell morphology which, if lost, can results in RPE cell death and IRD [262,276,277,278,279,280].

In terms of lipid regulation, the RPE expresses multiple genes that act to oxidize retinal lipids or otherwise alter their structures. CYP4V2, a major cytochrome in present in ARPE-19 cells, is a principal metabolizer of polyunsaturated fatty acids, such as docosahexaenoic acid and eicosapentaenoic acid, which are disc membrane lipids internalized by RPE cells during phagocytosis of the outer segments. The accumulation of these lipids is a primary characteristic of Bietti’s crystalline corneoretinal dystrophy (BCD) and has been linked to mutations in *CYP4V2*, the gene encoding CYP4V2 [281,282]. Loss of this gene causes accumulation of these lipids and subsequent RPE cell death and IRD. Regarding lipid-associated RPE genes, *HADHA*, the gene encoding for hydroxyacyl-CoA dehydrogenase, is the alpha subunit of a trifunctional lipid oxidation enzyme complex that functions to catalyze the final steps in fatty acid oxidation [283]. Loss of this protein’s function was shown to lead to an increase in cytosolic lipids in RPE cells and a subsequent loss of tight junction integrity between cells, resulting in RPE cell death and IRD similar to CYP4V2 dysfunction [283,284,285,286,287].

## 5. Retinal and Photoreceptor Development, Metabolism, and Homeostasis

The retina is a highly ordered laminar structure containing a multitude of different cell types: neuronal, glial, and immune. As the retina develops in stages (ganglion cells forming first, followed by photoreceptors, and Müller glia forming last), any alterations in the gene expressions and timing of the process can cause deleterious effects, including retinal degeneration [288,289,290,291,292]. These players include transcription factors, enzymes, receptors, and other important physiological proteins.

### 5.1. Transcription Factors and Nuclear Proteins

A myriad of transcription factors (TFs) and nuclear proteins are necessary for the proper development and function of the retina. Many of these TFs are responsible for determining the cell fate of retinal progenitor cells. There are four TFs with known IRD-associations: *CRX*, *NR2E3*, *NRL,* and *OTX2.* These code for cone rod homeobox (CRX), nuclear receptor subfamily 2 group E member 3 (NR2E3), neural retina leucine zipper (NRL), and orthodenticle homeobox 2 (OTX2), respectively. CRX, NR2E3, and NRL promote the expression of photoreceptor-specific genes such as *RHO*, *GNAT2*, and *OPN1SW,* without which, one or more photoreceptor types will not develop or develop abnormally and cause an IRD [293,294]. This is especially notable in the rod-dominated retinas of mammals where a loss of CRX or NRL leads to IRD due to the loss of all rod cells [293,294,295]. CRX is also responsible for expression of the RP gene *FAM161A,* mentioned previously [98]. OTX2 is responsible for expression of multiple neuronal and ocular developmental genes, including *RAX*, *PAX6*, *HES1*, and *SIX3*, making its loss through mutation detrimental to proper development of the eye as a whole but especially outer retinal components, including photoreceptors, RPE, and Müller cells [296,297,298]. Another transcription factor associated with IRD is *ZNF408* which codes for zinc finger protein 408 (ZNF408). ZNF408 has been shown to be associated with exudative vitreoretinopathy due to its role in the developing vasculature of the retina [299,300]. Other genes coding for nuclear proteins associated with IRDs that are not TFs include *SAMD11* coding for sterile alpha motif (SAM) domain protein 11 (SAMD11), *ATXN7* coding for ataxin-7 (ATXN7), and *CHD7* coding for chromatin helicase DNA-binding protein 7 (CHD7). SAMD11 is known to play a role in the polycomb repressive complex 1, a complex required for the proper development of rod cells [301]. ATXN7 which causes spinocerebellar ataxia type 7, a neurodegenerative disease, results in the death of rod cells as well, likely through the downregulation of rod cell proteins, including those for phototransduction and development [302,303,304]. Lastly, CHD7, a chromatin remodeling protein that exposes genes for transcription, has been shown to cause CHARGE syndrome, which has an associated retinopathy and is, in part, attributed to the loss of the genes downstream of CHD7′s function (i.e., genes located in the remodeled chromatin) [305,306,307].

### 5.2. Transcriptional and Translational Regulation

As one would expect, loss of proper regulation of transcriptional and translational activity can lead to highly deleterious results as these processes are of the utmost importance for cell survival. This is no different in the retina. Currently, eight genes associated with transcriptional regulation are known to be causal in IRDs. These genes, *PRPF3*, *PRPF4, PRPF6, PRPF8*, *PRPF31*, *SNRNP200*, *RP9* (or *PAP-1*), and *CWC27* code for the proteins pre-mRNA processing factors 3, 4, 6, 8, and 31, small nuclear ribonucleoprotein U5 subunit 200, PIM-1 associated protein 1, and CWC27 spliceosome- associated cyclophilin. These proteins are all essential for the correct post-transcriptional processing of messenger RNAs, specifically splicing and spliceosome formation [308,309,310,311,312,313,314,315,316,317,318,319,320,321,322]. Without proper functioning of these spliceosome components, many of the genes associated with photoreceptor function and health are not properly expressed, including *RHO*, resulting in IRD [308,309,310,311,312,313,314,315,316,317,318,319,320,321,322,323,324,325]. In terms of translation, the gene *OAT* is also associated with IRD. This gene codes for ornithine δ-aminotransferase, the enzyme responsible for conversion of the amino acid ornithine to the amino acid proline. Mutations in this gene result in two deleterious events where the cell loses its ability to generate proline while also accumulating ornithine, causing degeneration in the form of gyrate atrophy of the choroid and retina [326,327,328]. Retina and anterior neural folding homeobox 2 (RAX2) acts as a transcriptional coactivator in that it cooperatively interacts with CRX to trans activate photoreceptor-specific genes [329].Alternative splicing variant mutations in Rax2, the gene encoding RAX2, have been associated with cone-rod dystrophy type 11 and RP [330]. Interestingly, the gene *CERKL*, previously associated with outer segment phagocytosis, has also been associated with translational regulation through interactions with components eIF3B and PABP, indicating this role could also have an effect on photoreceptor function through polysome formation, a process necessary for production of many overly expressed proteins such as RHO [331].

### 5.3. Retinal Architecture: Cell Adhesion, Extracellular Matrix (ECM), and Basement Membranes (BMs)

*CRB1*, a major gene involved in retinal development and homeostasis, especially in photoreceptors and Müller glia, is crumbs homolog 1 (CRB1). CRB1 is mainly localized apically in the retina and is necessary to maintain retinal polarity and helps to sustain proper cell-to-cell contacts between different cell types, including Müller cells, photoreceptors, and RPE. Without these, the intercellular connections of photoreceptors and Müller glia begin to degrade, and the cells are no longer able to properly interact and function. The loss of these cell-to-cell junctions results in decreased retinal integrity and accumulation of toxic photoreceptor products in the outer retina and subsequent degeneration of photoreceptors leading to the clinical presentation of IRD [1,332,333,334,335]. While CRB1 is a major protein involved in retinal integrity, other genes are also involved in creating and maintaining cell-to-cell adhesions and when mutated cause IRDs. *CDHR1* and *CDH3* are all genes coding for cell adhesion proteins responsible for different cellular connections in the retina. Cadherin-related family member 1 (CDHR1), encoded by *CDHR1,* plays a key role in maintaining the junction of the inner and outer segments of photoreceptors, maintaining their structure and integrity [186,336,337,338]. Cadherin 3 (CDH3), encoded by *CDH3,* has been localized to the cell-to cell junctions between RPE monolayer cells and is likely part of the blood-retinal barrier created by the RPE [186,338,339]. The last gene thought to affect cell-to-cell adhesion is *C1QTNF5*, which codes for C1q-tumor necrosis factor 5 (C1QTNF5), a secretory and membrane-associated protein. Mutation in *C1QTNF5* have been identified to cause late-onset retinal macular degeneration (L-ORMD), characterized by lipid-rich deposits between the RPE and Bruch’s membrane. A Ser-to-Arg mutation at amino acid residue 163 (S163R) within C1QTNF5 has been proposed as the underlying cause in the proteins lack of adhesive function in creating cell-to-cell adhesions, resulting in deficient retinal integrity and homeostasis [186,280,338,340,341]. Interestingly, C1QTNF5 also appears to be a component of basement membranes (BMs), another critical aspect of retinal integrity and homeostasis discussed hereafter.

Another important aspect of the development and homeostasis of the retina is production and maintenance of the extracellular matrix (ECM) and, as stated previously, BMs. *IMPG1* and *IMPG2* code for interphotoreceptor matrix proteoglycan 1 and 2 (IMPG1 and IMPG2, respectively). These proteins are contained within the retina and are necessary for the formation and maintenance of the ECM around the photoreceptors themselves and between the RPE and photoreceptors. IMPG2 is thought to be necessary for proper localization of IMPG1, meaning genetic mutations in either will result in loss of the proper ECM formation and retinal degeneration [342,343,344,345]. Abnormalities in this gene have also been shown to lead to toxic accumulations in the subretinal space that result in photoreceptor degeneration [342]. Mutations in genes responsible for maintaining and altering the ECM, *TIMP3* and *ADAM9*, are both known as causal genes for IRD [346,347,348,349,350]. These genes code for tissue inhibitor of metalloproteinase 3 (TIMP3) and A disintegrin and metalloproteinase domain-containing protein 9 (ADAM9), respectively. ADAM9 is a matrix metalloproteinase that breaks down and remodels the ECM and may have a role in the outer segment phagocytosis of the RPE. Inversely, TIMP3 functions as an inhibitory protein acting to halt enzymes that function to degrade matrix components, such as metalloproteinases. Mutations in *ADAM9* have been found to cause CRD, while mutations in *TIMP3* have been shown to cause RP due to disrupted ECM remodeling [346,347,348,349,350]. In terms of components of BMs, certain collagen genes are known to associate with IRDs. *COL2A1* coding for collagen type II α1 (COL2A1) is a component of BMs and the vitreous humor. Mutations in *COL2A1* result in dysregulated fibrillar lamellar development of the vitreous humor, as well as abnormal collagen helices. Mutations in *COL2A1* have been linked to causing Wagner syndrome, characterized as an autosomal-dominant vitreoretinopathy [282]. In addition, *COL4A5*, another collagen-coding gene, results in an X-linked maculopathy when mutated due to a similar loss of BM integrity [351]. Finally, EGF containing fibulin extracellular matrix protein 1 (EFEMP1) is a glycoprotein that plays a role in extracellular matrix organization, primarily in Bruch’s membrane [352,353]. An SSCP analysis study identified a potential disease-causing exon shift in *EFEMP1*, the gene encoding EFEMP1, in patients with both Doyne Honeycomb and Malaria Leventinese Retinal Dystrophy [354]. Mutation EFEMP1 have been associated with the development of drusen deposits and retinal neovascularization, two primary characteristics identified in both Doyne Honeycomb Retinal Dystrophy and Malaria Leventinese [352,353,354].

### 5.4. Retinal Energetic Metabolism

A cell’s ability to produce and use its own energy is a critical aspect of cellular biology. Of all the tissues in the body, the retina is one of, if not the highest consumers of oxygen and ATP [355]. As such, any moderate loss of energetic metabolism could result in loss-of-function of many of the processes occurring during phototransduction, the visual cycle, or any number of the pathways associated with retinal integrity and homeostasis. Thus, mutations in genes associated with glycolytic or mitochondrial function could result in an IRD. Alternatively, loss of proper energetic metabolism will typically also have a deleterious effect on other highly metabolically active tissues. This is the case of known IRD-associated gene mutations involved in energetic metabolism.

*CERKL*, a gene previously mentioned to influence photoreceptor outer segment turnover, has also been identified to impact energetic metabolism when mutated. The gene product, ceramide kinase-like protein (CERKL), shown to additionally localize to mitochondria of retinal ganglion cells, functions in the regulation of mitochondrial metabolism in the retina via dysfunctional mitochondrial bioenergetics and altered mitochondrial distribution, seen to cause RP [356,357,358]. Additionally, CERKL shows protective function for the photoreceptors against oxidative stress through the antioxidant pathway [357]. A second gene implicated as important in cellular respiration is *NMNAT1*, encoding for nicotinamide mononucleotide adenylyltransferase 1 (NMNAT1). Proper NMNAT1 function is necessary for nuclear NAD+ biosynthesis within all nucleated cells, meaning loss of this enzyme would result in a depletion of primary electron carriers in glycolysis and oxidative phosphorylation [359,360]. Approximately 5% of LCA cases are attributed to mutations in *NMNAT1* and are observed to cause elevated levels of oxidative stress and damaging effects leading to retinal degeneration [359,360,361]. In fact, in a recent study involving mutant NMNAT1 mice, the first cell type to show degeneration were photoreceptors, exemplifying how important proper functioning NAD+ biosynthesis is to cellular maintenance [361]. Another seemingly ubiquitously expressed gene implicated in IRDs is *HK1*. *HK1* encodes for the human hexokinase 1 (HK1), which is critical in the metabolism pathway of glycolysis, where it phosphorylates glucose to prevent it from diffusing out of the cell [362]. As the retina requires a high energy cost, mutations in HK1 have been shown to decrease energy utilization and lead to photoreceptor degeneration, especially in rods that heavily use glycolysis for their energy needs [362,363,364]. *PANK2*, the last gene discussed in respiration associated IRD, encodes pantothenate kinase 2 (PANK2). PANK2 is trafficked to the mitochondria, which indicates it is likely necessary within energy metabolism [365]. In fact, it is a necessary regulatory enzyme with the synthesis of coenzyme A (CoA). As CoA, or its alkylated derivative acetyl-CoA, is necessary for all metabolic functions regardless of carbon source, the loss of its proper function in photoreceptor cells would result in dire consequences, including the loss of energy production needed to maintain the function and production of fatty acids [365,366].

### 5.5. Cell Signaling, Iron Regulation, Autophagy, and Peroxisome Activity

Several signaling receptors, iron regulatory proteins, and autophagy-related or peroxisome-related proteins are necessary for development and homeostasis and are associated with IRD. Jagged-1 (JAG1), known for its role as a ligand in NOTCH signaling pathways, including the proper development of multiple sensory cell types and angiogenesis via JAG1-NOTCH interactions causing targeted gene transcription. Mutations in *JAG1* have been associated with familial exudative vitreoretinopathy (FEVR) and is characterized by incomplete vascularization of the peripheral retina [186,367,368,369,370]. Similarly, frizzled-4 (FZD4) acts as a receptor downstream of Norrin/Wnt signaling and is involved in angiogenesis and retinal development [371,372]. Comparable to mutations in *JAG1*, mutations in *FZD4* (the gene encoding for FZD4) have also been linked to causing FEVR via its effects on dysregulated blood-retina-barrier integrity [371,372,373,374].

Iron regulation is a homoeostatic function of necessity as iron maintains a multitude of purposes, including heme production, enzyme cofactors, and transcriptional regulators. Unfortunately, iron can also act to a cell’s detriment by oxidizing cellular components and inducing iron overload. The gene *FLVCR1*, coding for feline leukemia virus subgroup C receptor 1 (FLVCR1), is a gene involved heavily in iron regulation and causes IRD when mutated [375,376,377]. The molecular mechanism underlying this is thought to be a loss of proper localization of the protein to the cell membrane where it works to transport iron in and out of the cell. Dysfunctions in the process have been seen to result in excessive iron within the cell and subsequent cell death [375,376,377].

In terms of autophagy and lysosomal deficiency, multiple mutated genes have been shown to cause IRDs. *CLN8* codes for ceroid lipofuscinosis type 8 (CLN8) which functions to transfer lysosomal enzymes that are necessary for lysosome biogenesis from the endoplasmic reticulum to the Golgi complex [378,379]. Without proper enzyme trafficking attributed to mutations in *CLN8*, there is a consequential lysosomal accumulation due to lack of enzymes for breakdown of lysosomal cargo and resultant cell death, including photoreceptors and RPE [379,380]. *CLN3*, another gene related to *CLN8* and lysosomes, codes for ceroid lipofuscinosis type 3 (CLN3). Although the function of CLN3 remains unclear, it has been suspected to play a role in clearance of glycerophosphodiesters from lysosomes. In recent studies generating a *CLN3* gene deletion in a mouse model, researchers noted RPE atrophy and degeneration, followed by significant vision impairment. It is likely the loss of CLN3 induces lysosomal storage disease and subsequent cell death. It was also proposed that in the RPE, the lack of functioning CLN3 induces a decrease in mitochondrial function with an increase in autophagy, likely degrading mitochondria and causing cell death [381,382]. Another gene familiar to this review and associated with autophagy regulation is *CERKL*. In addition to its other functions mentioned, CERKL has also been shown to regulate autophagy via its regulation of sirtuin-1 (SIRT1), a deacetylase of components essential to proper autophagy. Dysregulations in autophagy caused by mutations in *CERKL* have been seen to negatively impact the photoreceptor outer segment phagocytosis identified in RP [357,383]. The two final genes known to cause IRD through lysosomal dysfunction are *HGSNAT* and *VSP13B*. *HGSNAT* encodes heparan-α-glucosaminide-*N*-acetyltransferase (HGSNAT), a lysosomal acetyl transferase that functions to catalyze the transmembrane acetylation of heparan sulfate in BMs and allows for the step-wise breakdown of glycosaminoglycans (GAGs). Mutations in *HGSNAT* induce lysosomal storage disorder as unacetylated heparan sulfate is toxic to cells and accumulates in lysosomes, resulting in RP. *VPS13B* encodes an autophagy-associated protein called vacuolar protein sorting 13 homolog B (VPS13B). Mutations in *VPS13B* appear to cause increased lysosomal function, resulting in excessive degradation of cellular components [384,385,386,387,388]. Mutations in *VPS13B* have also been identified in many people with Cohen’s syndrome which is a rare autosomal recessive disorder characterized by a multitude of neurodevelopmental disorders, including retinal dystrophy [387].

The final genes to mention in relation to homeostasis are genes associated with the peroxisome and its functions. Acyl-CoA binding-domain-containing-5 (ACBD5), coded for by the *ACBD5* gene, is a gene critical for the peroxisomes long-chain fatty acid oxidation [389,390,391]. ACBD5 is proposed to be anchored in the peroxisomal membrane and a transporter of very long chain fatty acids (VLCFAs) into the peroxisome for oxidation [389,390,391]. Loss of this gene results in increased levels of VLCFA incorporation into phospholipids and a resultant cell death and retinal dystrophy [389,390,391]. Another IRD-associated gene involved in fatty acid metabolism and the peroxisome is *FALDH*. *FALDH* encodes for fatty aldehyde dehydrogenase (FALDH). The primary role of this protein within the retina is the oxidation of medium and long chain aldehyde derivatives preventing oxidative stress to the peroxisome and subsequent cellular damage [392,393]. When this gene and subsequent protein are dysfunctional, there is an accumulation of fatty alcohols and fatty aldehydes in the retina. When these accumulate, the cellular membrane integrity is compromised by lipid peroxidation leading to abnormal photoreceptor function [392,393,394,395].The last genes to mention in peroxisomal function are *PEX1* and *PEX6* which code for the proteins peroxisomal biogenesis factors 1 (PEX1) and 6 (PEX6), respectively. These proteins are localized to the cytoplasm but become anchored in the peroxisomal membrane during formation and are both required for the proper import of peroxisome-targeted biomolecules. The dysfunction of either protein results in a loss of peroxisome formation and function and ultimately cell death [396,397,398,399,400]. In recent studies, hypomorphic mutations in either *PEX1* or *PEX6* have been observed to cause Heimler syndrome, a recessively inherited disease causing multiple sensory and neurological defects, including visual [397,400,401].

## 6. Proinflammatory Signaling and Leukocyte Activation and Invasion

A final, and notably one of the most recent, mechanism of degeneration associated with IRDs is retinal inflammation, specifically the effects of damage-associated molecular patterns (DAMPs), proinflammatory cytokine release, and microglial/monocytic invasion of the retina [402,403,404,405,406,407,408,409,410,411]. While IRDs are initiated by a mutation in any one of the many genes listed above (as well as others not covered here), studies are now showing that the degenerative phenotype observed is progressed by the retina’s cellular and molecular responses to the mutation’s effects. Multiple aspects of retinal inflammation have been proposed to contribute to the photoreceptor loss and these pathways have been shown in animal models, humans, or both [4,15,403,404,405,408,409,412,413,414,415,416]. As the photoreceptors fall victim to the disrupted pathways induced by the IRD mutation, their stress signals and initiation of apoptosis induces responses in multiple cell types, including Müller glia and the resident macrophages of the retina, microglia. DAMPs sensed by these cell types induce cytokine release, especially by microglia, which become M1 activated and release multiple cytokines, including interleukins (IL-1β, IL-6, and IL-18) and tumor necrosis factor alpha (TNFα) [4,15,403,404,405,409,412,413,414,415,417]. These cytokines activate multiple pathways in retinal cells, including the Janus kinase/Signal transducer and activator of transcription (JAK/STAT) pathway; nuclear factor-kappa B (NF-κB) activation; and induction of the NOD-, LRR-, and pyrin domain-containing protein 3 (NLRP3) inflammasome [15,412,414,418]. A particularly interesting aspect of these pathways is a possible positive feedback loop inducing further exacerbation of the M1 microglial activation, causing overly phagocytic microglia and, as shown recently, invading monocytes from circulation to perform a process called phagoptosis [15,408,410,411,414,419,420]. Phagoptosis, discovered in the last decade, is a form of cell death instigated by phagocytosis of non-apoptosing, still viable cells. This exacerbated M1 state also causes further release of cytokines that induce cell death of photoreceptors, including TNFα and IL-1β. These cytokines and others such as IL-6 can induce the activation of STAT3, NF-κB, and the NLRP3 inflammasome in other cell types, including RPE and downstream neurons of the retina [4,15,414,421]. What results is enhanced photoreceptor loss due to mutation-induced apoptosis, cytokine-induced apoptosis, and macrophage-induced phagoptosis advancing the degeneration and rate of vision loss. As a result, treatments specifically targeting these inflammatory pathways and cells have become a hot topic area of research in recent years [4,15,404,405,407,409,413,414,415,416,422].

## 7. Concluding Remarks

IRDs, while rarer than other more prominent blinding diseases such as glaucoma and age-related macular degeneration, exhibit much more severe and early-onset degenerative phenotypes progressing much more rapidly often with little to no window for treatment. In this review, we delved into myriad pathogenetic molecular mechanisms, which all result in a similar outcome—complete loss of photoreceptors and blindness. Gene therapies for these congenital disorders are slow to produce and lack expedience when presented for FDA approval, especially when these treatments must be applied to children in cases such as LCA, Stargardt’s disease, and severe RP, where the disease symptoms begin in childhood and leave them blind by adolescence. Due to this unfortunate situation, therapies aimed at slowing the rate of photoreceptor degeneration and preserving sight for as long as possible are coveted in the realm of IRDs. Interestingly, our lab and others have shown that chronic inflammation and overactivation of macrophages in the outer retina progresses the rate of degeneration in mouse models of IRDs [4,15,403,404,407,408,414,416,420,423]. It is to this end that studies aimed at targeting the macrophages and/or cytokines associated with the proinflammatory environment are underway to serve this unmet need in the vision community with the hope to preserve the retinas of these burdened patients while cures are being developed.

## Figures and Tables

**Figure 1 biomolecules-13-00271-f001:**
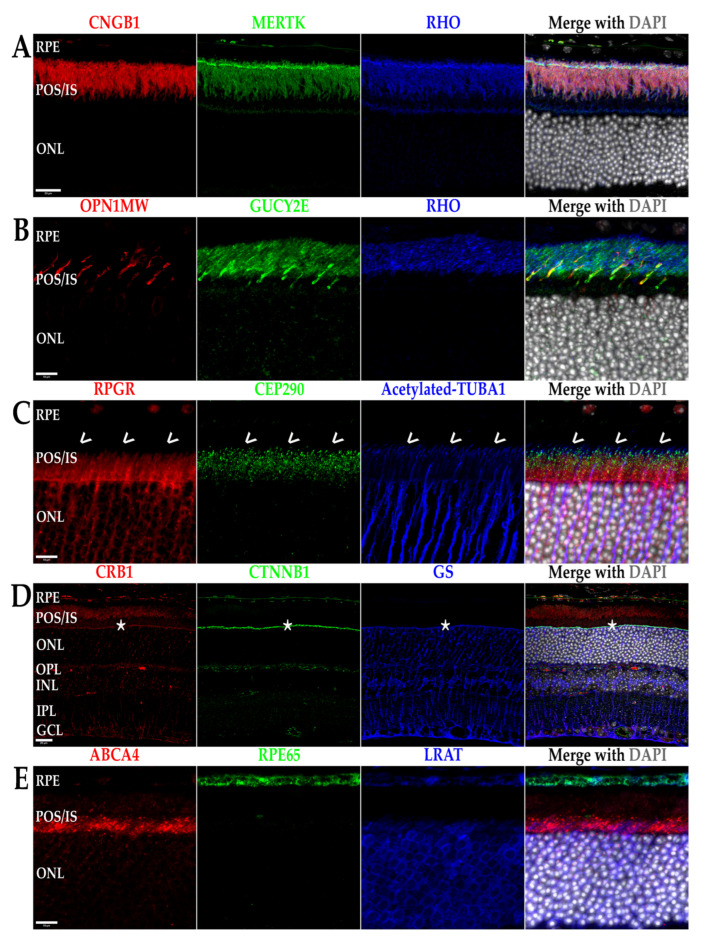
Immunohistochemical localization of many common inherited retinal dystrophy (IRD)-associated proteins. (**A**,**B**) Outer segment (POS) localization of phototransduction proteins CNGB1 (**A**, red), OPN1MW (**B**, red), GUCY2E (**B**, green), and RHO (**A**,**B**, blue), as well as phagocytosis protein MERTK (**A**, green). (**C**) Connecting cilium (>) localization of proteins RPGR (red), CEP290 (green), and acetylated TUBA1, a marker of the axoneme (blue). (**D**) Outer limiting membrane (*) localization of proteins CRB1 (red), CTNNB1 (β-catenin, green), a marker of cadherin junctions, and GS (glutamine synthetase, blue), a marker of Müller glia. (**E**) Photoreceptor localization of ABCA4 (red) and LRAT (blue), and retinal pigment epithelial (RPE) localization of RPE65 (green) and LRAT (blue). Nuclei labeled with DAPI (grey). RPE, retinal pigment epithelium; POS/IS, photoreceptor outer segments/inner segments; ONL, outer nuclear layer; OPL, outer plexiform layer; INL, inner nuclear layer; IPL, inner plexiform layer; GCL, ganglion cell layer. Scale bars = 20 μm (**A**,**D**) and 10 μm (**B**,**C**,**E**).

**Figure 2 biomolecules-13-00271-f002:**
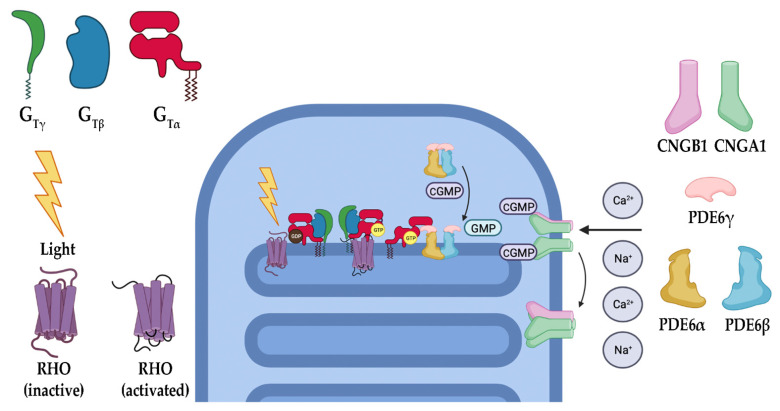
Diagram of phototransduction activation pathway proteins. Upon activation of RHO by absorption of a photon of light by its chromophore 11-*cis* retinal, RHO shifts its helical structure to allow for activation of the G-protein transducin (G_T_). G_T_, composed of α, β, and γ subunits, exchanges a GTP for a GDP in its activation site, allowing for separation of the α subunit from the βγ subunit complex, which then goes to activate the PDE6 enzyme by releasing the inhibition of its γ subunit. PDE6 then hydrolyzes cGMP to GMP, causing closure of the CNG channels, composed of three α and one β subunit, and subsequent hyperpolarization of the photoreceptor. This slows glutamate release at the synapse and propagates the electrical signal to the second and third-order neurons of the retina, the bipolar cells, and ganglion cells, respectively.

**Figure 3 biomolecules-13-00271-f003:**
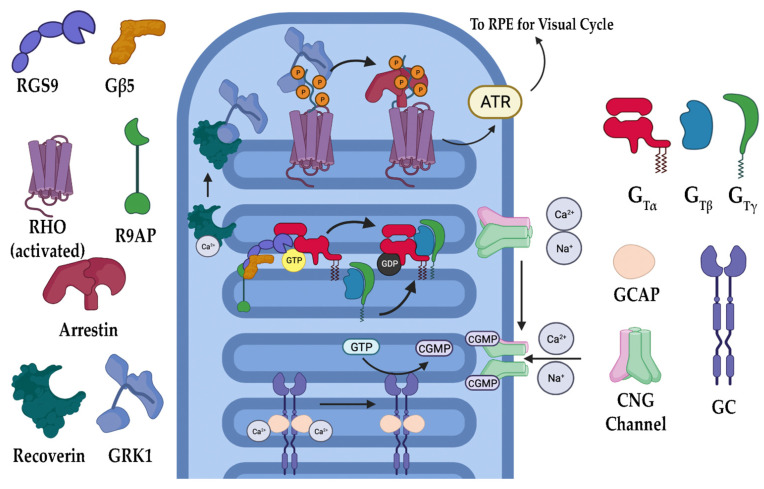
Diagram of phototransduction inactivation pathway proteins. After RHO activation and the subsequent cascade, the resulting decrease in calcium due to CNG channel closure releases the inhibition of recoverin, allowing for it to interact with and activate GRK1. GRK1 then phosphorylates RHO on its carboxyl terminus up to 6 times on serine and threonine residues, inducing RHO capping by arrestin. The isomerized all-*trans* retinal chromophore (ATR) is then hydrolyzed from the opsin protein and sent to the RPE for *cis* isomerization. G_T_ has an intrinsic GTPase activity that is further enhanced by the trio of RGS9, Gβ5, and R9AP causing hydrolysis of the γ phosphate of GTP forming GDP and inactivating G_Tα_, allowing for the β and γ subunits to reassociate with the α subunit. Along with recoverin, decreased calcium levels from CNG channel closure also release inhibition on GCAPs, allowing for them to activate GCs to produce cGMP from GTP for reopening of the CNG channels and depolarization of the cell to the dark state.

**Figure 4 biomolecules-13-00271-f004:**
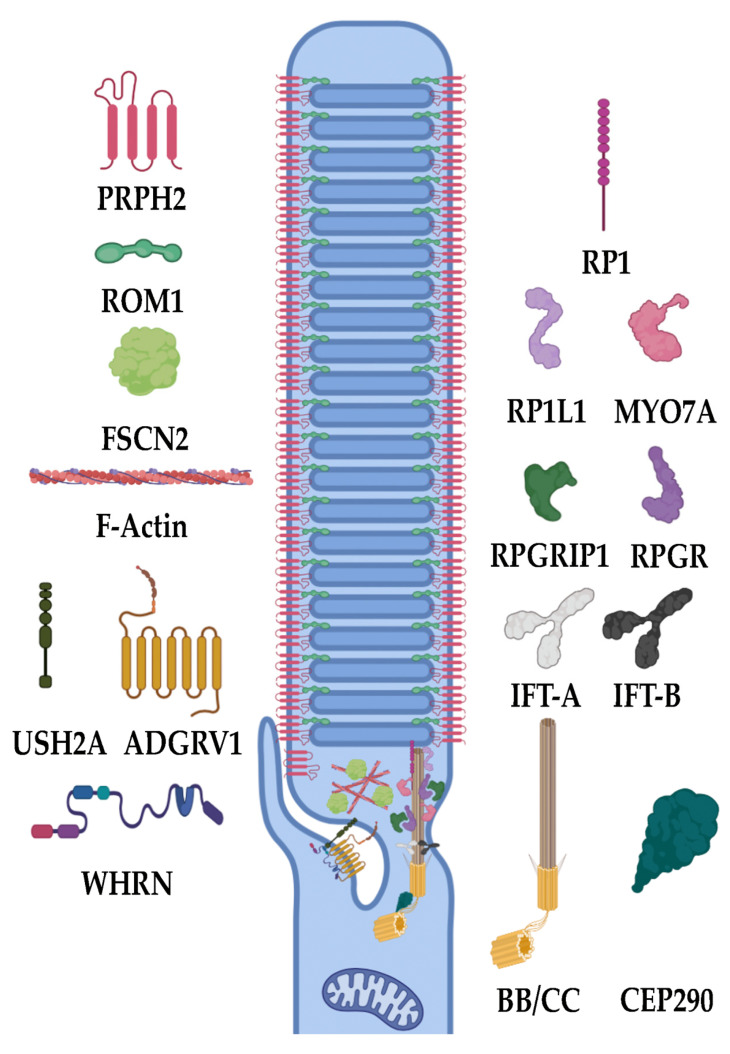
Diagram of ciliary and disc formation proteins. The connecting cilium (CC) forms from the basal body (BB), where CEP290 assists in tubulin nucleation. Trafficking protein complexes MYO7A, IFT-A, and IFT-B work to traffic important ciliary proteins and cargo anterograde and retrograde. Usher syndrome proteins USH2A, ADGRV1, and WHRN synergistically work to tether the periciliary ridge to the base of the outer segment. Ciliary proteins RPGR and RPGRIP1 work in concert to stabilize the CC, while RP1 and RP1L1 work to initiate the process of disc formation in the outer segment. Actin bundling protein FSCN2 works at the base of the outer segment to form F-actin bundles for disc formation to ensue by PRPH2 and ROM1, two of the major disc formation proteins tasked with flattening the circular vesicles into discs, a less energetically favorable configuration.

**Figure 5 biomolecules-13-00271-f005:**
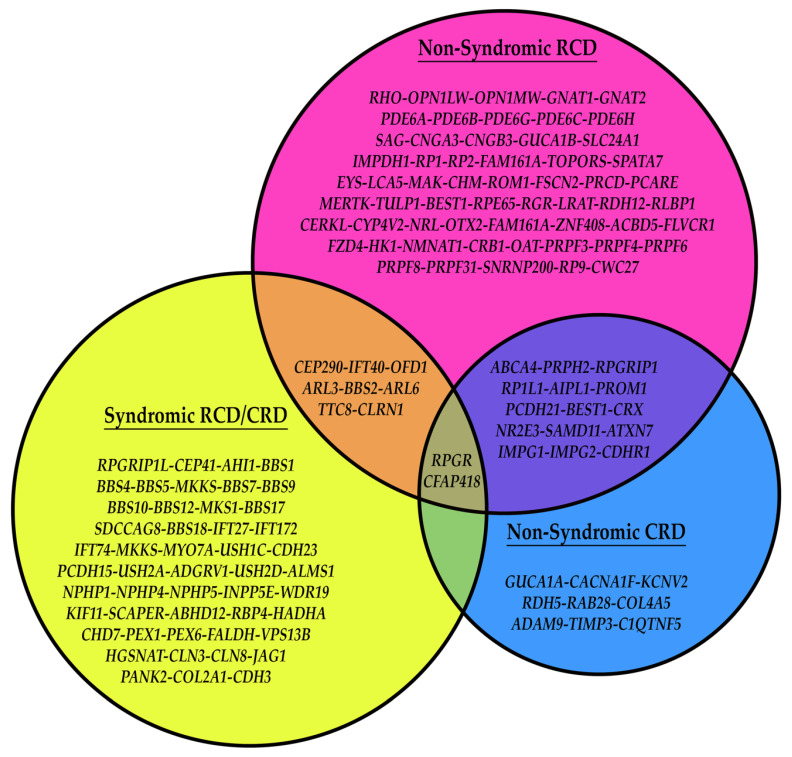
Venn diagram representation of the overlap of genes between different forms of inherited retinal dystrophies (IRDs). While hundreds of genes may cause a form of IRD, including non-syndromic rod-cone dystrophies (RCDs, magenta), non-syndromic cone-rod dystrophies (CRDs, blue), or syndromic RCDs/CRDs (yellow), several genes can result in more than one form when mutated (non-syndromic RCDs and syndromic RCDs/CRDs, orange; non-syndromic CRDs and syndromic RCDs/CRDs, purple; non-syndromic RCDs, non-syndromic CRDs and syndromic RCDs/CRDs, light brown), all dependent on the mutation and its effects. Note the overlap between the different groups consists of primarily ciliopathy genes among non-syndromic RCDs/CRDs and syndromic RCDs/CRDs and retinal function and development genes overlapping between the non-syndromic RCDs and non-syndromic CRDs.

**Figure 6 biomolecules-13-00271-f006:**
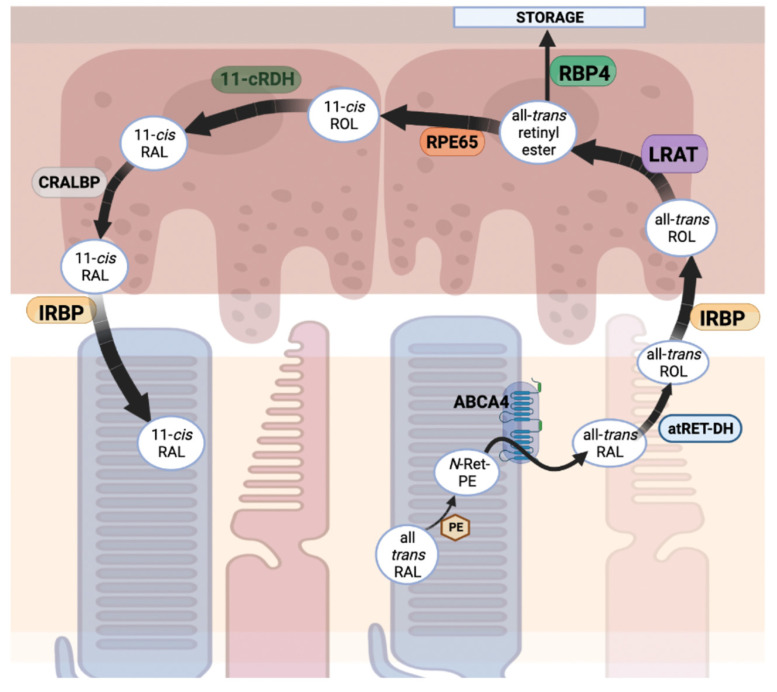
Diagram of the visual cycle proteins. The primary role of the visual cycle is the replenishment of 11-*cis* retinal (11-*cis* RAL) in photoreceptor cells via the re-isomerization of all-*trans* retinal (all-*trans* RAL). Once the all-*trans* retinal is cleaved from rhodopsin during phototransduction, it reacts with phosphatidylethanolamine (PE) to form *N*-retinylidene-phosphatidylethanolamine (*N*-Ret-PE), which binds the flippase ABCA4, resulting in a confirmational change that releases all-*trans* retinal into the cytoplasmic leaflet of the disc membrane. All-*trans* retinal is then reduced by all-*trans* retinol dehydrogenase (atRET-DH), forming all-*trans* retinol (all-*trans* ROL). All-*trans* retinol is trafficked by the retinal-binding protein IRBP from the photoreceptor outer segment to the RPE. Once in the RPE, the acyl transferase LRAT reacts an acyl group with all-*trans* retinol to form an all-*trans* retinyl ester. All-*trans* retinyl esters are a nontoxic storage form of retinal, which may be trafficked for storage by another retinal-binding protein RBP4. The rate-limiting step of the visual cycle is the isomerization of the all-*trans* retinyl ester to form 11 *cis*-retinol (11-*cis* ROL), in which 11-*cis* retinol dehydrogenase (11-cRDH) then oxidizes 11-*cis* retinol to form 11-*cis* retinal. 11-*cis* retinal is then chaperoned by the retinal-binding protein CRALBP to protect it from further isomerization or esterification. Finally, 11-*cis* retinal is trafficked from the RPE to the outer segment of photoreceptor discs, where it may bind opsin to regenerate functional rhodopsin.

**Table 1 biomolecules-13-00271-t001:** Antibodies used for Western blots and fIHC labeling of retinal sections.

Antibody Target	Host Species & IgG Isoform	Dilution	Source
ABCA4	Mouse IgG_1_	1:200	EMDMillipore
CEP290	Rabbit IgG	1:200	EMDMillipore
CNGB1	Mouse IgG_2a_	1:250	EMDMillipore
CRB1	Rabbit IgG	1:100	Boster Bio. Tech.
CTNNB1	Mouse IgG_2b_	1:200	Origene
GS	Mouse IgG_2a_	1:500	BD Trans. Lab.
GUCY2E	Rabbit IgG	1:100	FabGennix
LRAT	Rabbit IgG	1:100	ThermoFisher
MERTK	Rat IgG_2a_	1:100	ThermoFisher
OPN1MW	Rabbit IgG	1:100	Novus Bio.
RHO	Mouse IgG_1_	1:500	Santa Cruz Biotech.
RPE65	Mouse IgG_1_	1:200	ThermoFisher
RPGR	Rabbit IgG	1:100	abcam
TUBA1, Acetyl	Mouse IgG_2b_	1:250	EMDMillipore

## Data Availability

No new data were created or analyzed in this study. Data sharing is not applicable to this article.

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
