# Peer review of "Cellular and Molecular Mechanisms of Pathogenesis Underlying Inherited Retinal Dystrophies"

_biomolecules, 2023, doi:10.3390/biom13020271_

Round 1

Reviewer 1 Report

This review article is a well-written, exhaustive account of proteins and mutations involved in inherited retinal dystrophies, which eventually lead to blindness. This review covers a wide range of proteins and pathways involved in the proper function of photoreceptors and will be of general interest to scientists studying different aspects of photoreceptor function and homeostasis. Considering the vast number of proteins involved, in some places, it's difficult to follow the article which is understandable. The authors may consider including figures for some of the pathways discussed like phototransduction, ciliary trafficking, and/or visual cycle. My main concern is regarding the lack of experimental details for figure 1. Although it's a review article, since original data is presented, it should accompany the relevant experimental details."

Author Response

We thank the reviewer for their response and concerns. As requested, new figures have been included showing phototransduction proteins and the pathways, the visual cycle, and ciliary and disc formation. We were unable to include every single protein mentioned in the manuscript as this would have been too cluttered to discern much. As such, we included some of the key players. We have also included a methods section as requested and a list of antibodies used in Table 1.

Reviewer 2 Report

This is a comprehensive and well written review providing brief information about the large number of genes that have been implicated in inherited retinal dystrophies. I enjoyed reading it and only have a few comments or suggestions, as follows:

In the introduction, the authors describe the main causes of IRDs as cone-rod dystrophies or rod-cone dystrophies, but neglect to mention the contribution of primary RPE dysfunction (this is mentioned in the abstract and described in more detail in a subsequent section of the manuscript, but for clarity, should be added to the introduction as well).

No details of methods/antibodies used for staining shown in Figure 1, or which strain, age, sex, genotype of mouse was used for the retinal sections. Was this new data generated for this review article or previously published? Do human proteins share the same localisation?

There seem to be some omissions firstly in genes covered e.g. no mention of EFEMP1 or RAX2. How were the genes included in the manuscript identified –through a literature search or database search, and can details be added to the manuscript. A quick glance through self-citations suggests that publications with an author included on this manuscript are used in some instances in referencing when citation of an initial report may be missing e.g. mutated gene for C1QTNF5 in L-ORMD (Hayward et al 2003). 

I would like to see increased detail of the type of mutation in each gene and whether PTC/Frameshift/splicing/missense/null mutations lead to different phenotypes and how cellular and mouse models have been used to study this further (mouse models in particular implicated as crucial to understanding of disease mechanisms by concluding remarks, but there is little detail provided throughout the manuscript despite extensive referencing of such studies). The concept of IRDs with distinct causes converging on shared inflammatory pathways which could allow a common, gene-agnostic treatment is also worthy of expanding on as it will be of particular interest to Biomolecules readers.

Author Response

We thank the reviewer for their comments and criticisms. Please find the replies below:

1) RPE dysfunction is now mentioned in the introduction.

2) A methods section and table of antibodies used for Figure 1 is now included.

3)The genes suggested EFEMP1 and RAX2 have now been included. We agreee with the reviewer that there are genes not mentioned in the manuscript. We would have loved to include every gene now implicated in IRDs; however, this most assuredly would have made this review significantly longer than it already is and we took it upon ourselves to limit the genes covered. The new citation was also added. We had no intent of only self citing and appreciate the reviewers catch of this.

4) Some details of different forms of mutations in some of the genes, namely those that cause both syndromic and non-syndromic disease have been included and a short paragraph describing how different mutation types in the same gene can cause different disease phenotypes and subtypes.

5) We have also added a section on inflammation and expounded upon its role as a mechanism of disease pathogenesis.